# A rapid inducible RNA decay system reveals fast mRNA decay in P-bodies

Lauren A. Blake[1,2], Leslie Watkins[1,2], Yang Liu[1,2,5], Takanari Inoue[2,3] & Bin Wu[1,2,4] ✉

RNA decay is vital for regulating mRNA abundance and gene expression. Existing technologies lack the spatiotemporal precision or transcript specificity to capture the stochastic and transient decay process. We devise a general strategy to inducibly recruit protein factors to modulate target RNA metabolism. Specifically, we introduce a Rapid Inducible Decay of RNA (RIDR) technology to degrade target mRNAs within minutes. The fast and synchronous induction enables direct visualization of mRNA decay dynamics in cells. Applying RIDR to endogenous *ACTB* mRNA reveals rapid formation and dissolution of RNA granules in pre-existing P-bodies. Time-resolved RNA distribution measurements demonstrate rapid RNA decay inside P-bodies, which is further supported by knocking down P-body constituent proteins. Light and oxidative stress modulate P-body behavior, potentially reconciling the contradictory literature about P-body function. This study reveals compartmentalized RNA decay kinetics, establishing RIDR as a pivotal tool for exploring the spatiotemporal RNA metabolism in cells.

RNA is the essential intermediate biomolecule that transmits genetic information encoded in DNA into functional proteins. It is tightly regulated both at its birth (transcription) and death (degradation). RNA degradation is essential to maintain transcript homeostasis and clear defective RNA species. In eukaryotic cells, normal mRNA degradation is initiated by deprotection of its ends with deadenylation or decapping, followed by 5′ ⟶ 3′ degradation by XRN1 or 3′ ⟶ 5′ decay by the RNA exosome. Defective mRNAs are cleared by RNA quality control pathways; for instance, nonsense-mediated decay (NMD), no-go decay, or non-stop decay[1]. In the NMD pathway, a premature stop codon activates the RNA helicase UPF1, eventually committing the RNA to degradation by recruiting heterodimer SMG5/SMG7 to decay mRNA through the deadenylation/decapping pathway or the endonuclease SMG6 to cleave the mRNA[2].

RNA-containing membraneless organelles, including nuclear speckles, stress granules (SGs), and processing-bodies (P-bodies), play important roles in RNA metabolism, including splicing, modification, storage, and decay[3,4]. P-bodies were initially discovered in yeast and were shown to be enriched with RNA decay machineries, but devoid of ribosomes and translation factors[5,6]. Therefore, it was hypothesized that P-bodies were the sites of RNA decay. However, subsequent studies have challenged this view as RNA degradation still occurs in the absence of visible P-bodies[7,8]. Recent research suggests that P-bodies might serve as storage sites for mRNAs that can be translated again upon exiting P-bodies[9,10]. Unlike SGs, P-bodies exist in steady-state physiological conditions, and recruiting RNA to P-bodies typically requires the application of stress, such as amino acid starvation or osmotic stress[11]. During stress, a large variety of mRNAs are recruited to P-bodies, which mediates the stress response and recovery. Currently, the physiological function of P-bodies in unstressed states remains elusive. To address these questions, it is crucial to investigate the localization and degradation of specific RNAs in P-bodies and the cytoplasm separately under physiological conditions.

[1]Department of Biophysics and Biophysical Chemistry, Johns Hopkins University School of Medicine, Baltimore, MD 21205, USA. [2]The Center for Cell Dynamics, Johns Hopkins University School of Medicine, Baltimore, MD 21205, USA. [3]Department of Cell Biology, Johns Hopkins University School of Medicine, Baltimore, MD 21205, USA. [4]The Solomon H Snyder Department of Neuroscience, Johns Hopkins University School of Medicine, Baltimore, MD 21205, USA. [5]Present address: Department of Biochemistry, University of Utah, Salt Lake City, UT 84112, USA. ✉e-mail: bwu20@jhmi.edu

Conventionally, RNA decay is measured in bulk experiments by harvesting RNA at different time points after inhibiting transcription or pulse labeling of new transcripts[12]. Cells are lysed, and if necessary, fractionated to enrich certain cellular compartments. As a result, the spatial information is lost, and the temporal resolution is limited. Fluorescence imaging tracks the RNA and organelles in real time, which allows for direct visualization of biological events in subcellular compartments with temporal resolution compatible for mRNA decay.

Imaging mRNA at the single molecule level in live cells is crucial for unraveling the mechanism of RNA synthesis, transport, translation, and degradation. Single-cell / single-molecule imaging technology has enabled the direct measurement of transcription dynamics at the single allele level[13]. However, the spatiotemporal dynamics of RNA decay in cells remain poorly understood, due to the transient nature of degradation and the disappearance of signal being frequently confounded by imaging artifacts, such as photobleaching, or diffusion out of focus or out of the field of view. While enzymatic degradation of RNA occurs in seconds to minutes, many mRNAs in mammalian cells take hours before they are committed to degradation. The rapid diffusion of mRNAs in cells makes it challenging to track and capture the infrequent and transient decay process. Modulating RNA decay on demand would be instrumental because it can synchronize the transient process[14]. While existing methods, like RNA interference, are convenient to knock down target genes, it takes many hours to exert effects - too slow for studying decay dynamics. Thus, a method for inducing rapid and synchronous decay of RNAs is highly desirable.

In this study, we established a rapid inducible RNA decay system by recruiting an RNA degradation factor on demand, and quantified RNA in subcellular compartments in single molecule resolution. We demonstrated that RIDR can knock down target mRNAs faster than standard small interfering RNA (siRNA). The rapid synchronous decay allowed us to study the function of membraneless organelles, such as P-bodies. By combining RIDR with genetic and pharmacological perturbations, we revealed the functional role of P-bodies in RNA decay.

## Results

### An inducible RNA decay system that is fast and specific

This method is motivated by RNA's natural propensity to assemble into ribonucleoprotein particles (RNPs). RNA is not a naked polymer of nucleotides; it associates with numerous protein factors, which ultimately determine its fate. The protein composition of RNPs is constantly remodeled during the lifecycle of an mRNA. By tethering specific RNA binding proteins to the target mRNAs, their fate can be artificially influenced. To modulate RNA decay on demand, we implemented a chemically inducible dimerization (CID) system to control RNA metabolism by fusing an RNA decay factor and a sequence-specific RNA binding protein to a CID pair (Fig. 1a). The CID pair we utilized consisted of the FK506 Binding Protein (FKBP) and the FKBP−Rapamycin Binding domain (FRB) that rapidly dimerize at low concentrations of rapamycin (Rapa)[15]. The FRB/FKBP CID system has been applied to control protein dimerization and many cellular functions before[16,17].

The tethered RNA decay factor should meet several criteria. First, it needs to be non-toxic when over-expressed. Second, it should be active only when proximal to the target RNA, with minimal off-target effects. Third, it should efficiently prompt RNA degradation upon tethering. One such promising candidate, SMG7, functions in the nonsense mediated mRNA decay (NMD) pathway. It is one of the last factors recruited in the NMD pathway once an mRNA is committed to decay irreversibly[18]. Previously, it was demonstrated that tethering of the C-terminus of SMG7 (SMG7C) directly to an mRNA decreases its half-life up to 3 times without any upstream NMD factors[19]. Another candidate is the endonuclease in the NMD pathway, SMG6. The catalytic PIN domain (SMG6PIN) can also degrade target RNA when tethered[20]. We compared the efficiency of SMG6 and SMG7

degradation of target mRNAs when directly tethered. We constructed reporters encoding fluorescent protein mCherry with different numbers of MS2 Binding Sites (MBS) in the 3' untranslated region (UTR). The plasmids mCherry-nxMBSv5 (where $n = 0, 1, 3, 6, 12, 24$)[21] were co-transfected into HEK293T cells with the tandem MS2 Coat Protein (tdMCP) that specifically binds MBS motifs[22]. The expression of mCherry was then measured by flow cytometry (Supplementary Fig. 1a). We directly fused HaloTag-tdMCP to either SMG7C or SMG6PIN and measured the knockdown efficiency relative to a negative control without any RNA decay factor (Supplementary Fig. 1b, c)[23]. SMG7C can degrade target RNA more efficiently than SMG6PIN, achieving 68% knockdown of mCherry protein with just 3x MBS, while SMG6PIN required 24 stem loops to achieve similar levels of knockdown (Supplementary Fig. 1c, d). Therefore, we concentrated on SMG7C in the following experiments.

Next, we employed an inducible tethering system by constructing a bicistronic vector with FRB fused to SMG7C, and FKBP fused to HaloTag-tdMCP, with the latter translated from an internal ribosome entry site (IRES) to facilitate co-expression. We named the construct Rapid Inducible Decay of RNA (RIDR) (Fig. 1b and Supplementary Fig. 1e). We then co-transfected RIDR and the mCherry-24xMBS (*mCherry-MBS*) plasmids into HEK293T cells. Flow cytometry results show that mCherry levels decreased 74% upon induction with Rapa (Fig. 1b, c). The knockdown is not due to translation repression from Rapa, as it is dependent on tethering of SMG7C by tdMCP (Fig. 1b, c). Not surprisingly, more MBS in target mRNAs improved the knockdown efficiency (Supplementary Fig. 1f, g). All experiments were performed using a standard final concentration of 100 nM Rapa, though titration of Rapa revealed that RIDR had the same knockdown efficiency at 10 nM Rapa, and the efficiency decreased at 1 nM (Supplementary Fig. 1h).

To assess the speed of RIDR, we measured the kinetics of the mRNA decay using single-molecule fluorescent in situ hybridization (smFISH) with probes targeting the MBS region. In U-2 Osteosarcoma (U-2 OS) cells stably expressing both the *mCherry-MBS* reporter RNA and the RIDR construct, > 90% of *mCherry-MBS* RNA disappeared upon induction by Rapa for 2 h (Fig. 1d, e), which was much faster than the gold standard RNA interference using mCherry siRNA treatment (Fig. 1f). Importantly, a nontargeting endogenous mRNA, *hPol2RA*, decayed similarly with or without Rapa, demonstrating the specificity of the RIDR system (Fig. 1g). To avoid confounding with newly synthesized mRNAs, we applied transcription inhibitor 5,6-dichloro-1-beta-D-ribofuranosylbenzimidazole (DRB) in all experiments (Fig. 1f, g).

### RIDR induces rapid decay of endogenously labeled mRNAs

To further examine the effectiveness of RIDR, we applied the tool to endogenous genes tagged with MBS. 24x MBS have previously been knocked into the 3'UTR of mouse β-actin (ACTB-MBS) at the endogenous loci without influence on its function[24]. We stably expressed the RIDR construct in mouse embryonic fibroblasts (MEF) extracted from the mouse, here forth referred to as ACTB-MBS MEF. The expression of *ACTB-MBS* mRNA is substantially higher than the *mCherry-MBS* reporter, but it can be knocked down equally fast: with 95% knockdown at 2 h post-induction with Rapa. This is much faster compared to siRNA treatment, which only achieved 24% knockdown in 2 h (Supplementary Fig. 2a–e). The decay of endogenous *mPolR2A* mRNA without MBS was not influenced by Rapa induction, demonstrating the specificity of RIDR (Supplementary Fig. 2f).

### RIDR induces RNA granules formed on pre-existing P-bodies

The rapid and synchronous RNA decay induced by RIDR allows observation of emergent phenomena that are obscured by stochastic decay of single mRNAs. To track the RNA decay in real time, we performed live-cell imaging of *ACTB-MBS* mRNAs using FKBP-HaloTag-

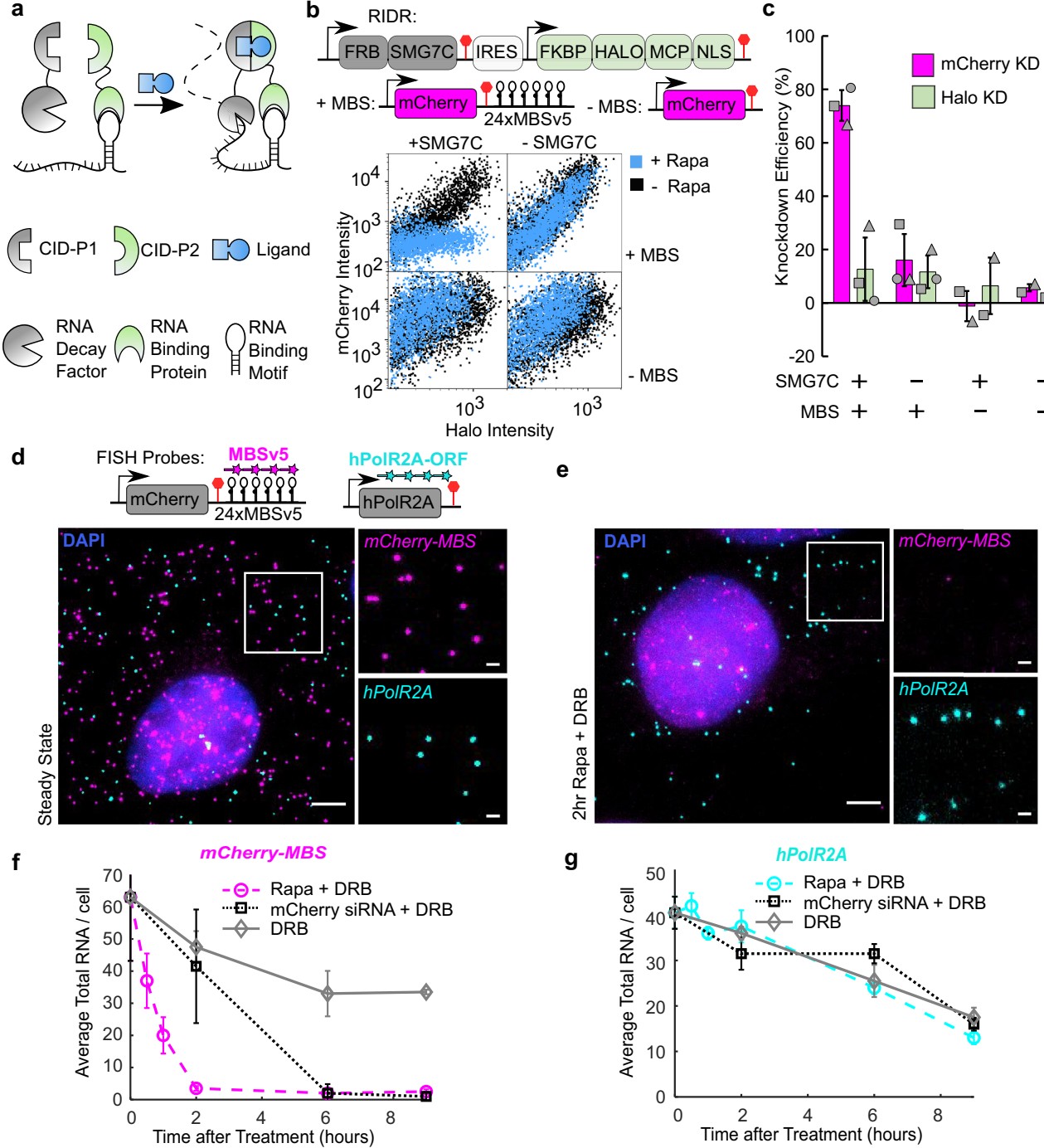

**Fig. 1 | RIDR system is fast, specific, and inducible. a** Schematic of a generalized inducible mRNA decay system for inducibly targeting an RNA binding motif with an RNA decay factor. **b** HEK293T cells were transiently transfected with mCherry-MBS and RIDR constructs. Cells were transfected for 12–16 h prior to preparation for flow cytometry. The cells were treated with 100 nM rapamycin (Blue) or DMSO control (Black) at the same time of transfection. FKBP-HaloTag-tdMCP in the RIDR construct was labelled with JF503-Halo-ligand. The fluorescence of single cells was measured by flow cytometry in mCherry and Halo503 and presented as a scatter plot. IRES: Internal Ribosome Entry Site. Raw flow cytometry data from +/− SMG7C and +/− MBS conditions from one replicate are presented. **c** Knockdown efficiencies of mCherry and HaloTag were quantified from flow cytometry experiments in (**b**) under conditions listed. The knockdown efficiency is calculated for each condition with respect to itself when Rapa is not added. The HaloTag signal is used as a negative control that does not depend on tethering and Rapa. 1000–3000 HaloTag-positive cells were quantified per condition. Data are presented as mean

values across 2, 3 biological replicates. Error bars represent the standard deviation. **d**, **e** Representative smFISH images of U-2 OS cells stably expressing *mCherry-MBS* and RIDR in steady state condition (**d**) and after 2 h of rapamycin treatment (**e**). The white box was enlarged on the right. *mCherry-MBS* FISH: magenta; *hPolR2A* FISH: cyan; DAPI: blue. Scale bars: 5 μm for original and 1 μm for zoomed images. **f**, **g** Quantification of time-resolved two-color smFISH experiment over 9 h after induction with Rapa (circle), mCherry siRNA (square), or DMSO (diamond). DRB was added in all experimental conditions to inhibit transcription. The number of transcripts for *mCherry-MBS* (**f**) and *hPolR2A* (**g**) were counted in the same cells for all time points. Rapa + DRB: circles; mCherry siRNA + DRB: squares; DRB alone: diamonds. Error bars represent standard deviation of the means of 2 biological replicates. 59–96 cells were quantified per condition across replicates (the precise number of cells per condition per replicate are given in the source data). Source data are provided as a Source Data file.

tdMCP labeled with Janelia Fluorophore (JFX646)[25]. We observed that RNA granules emerged within 5 min after induction and disappeared within 1 h (Supplementary Movie 1). To ascertain the identity of these granules, we simultaneously performed smFISH and immunofluorescence (smFISH-IF) with antibodies against common cytoplasmic RNA granules. First, we observed RIDR did not cause SG formation, as confirmed by staining with G3BP1, a common validated SG marker (Supplementary Fig. 3a, b). However, when Rapa was added to cells pre-treated with arsenite stress, some RNA granules were found touching SGs, but not colocalized with them (Supplementary Fig. 3c). Next, we checked if the RNA granules colocalized with P-bodies by staining with known P-body markers, decapping enzyme 1 A (DCP1a), DEAD box helicase 6 (DDX6), and 5′ ⟶ 3′ exonuclease XRN1. Though P-bodies are present in steady state conditions (Fig. 2a), the *ACTB-MBS* mRNA only colocalized with the DCP1a, DDX6, and XRN1 puncta after induction with Rapa (Fig. 2b and Supplementary Fig. 3d, e, respectively). At 2 h post-induction, a majority of *ACTB-MBS* mRNAs were depleted from both the cytoplasm and the P-bodies (Fig. 2c). In contrast, after treatment with siRNA against ACTB, no bright RNA granules formed that colocalized with P-bodies, even though individual *ACTB-MBS* mRNAs were found within P-bodies occasionally (Fig. 2d, e). The negative control *mPolR2A* mRNA did not accumulate in P-bodies in any treatment (Fig. 2f). The P-body number and average intensity did not change within the first hour of treatment. At longer time scales, the number of P-bodies decreased, and their individual average intensities increased (Fig. 2g, h), possibly due to merging of P-bodies at later time points. This is unlikely due to RIDR induction because the P-bodies' characteristics are similar in all conditions (Fig. 2g, h).

It was observed that in yeast, fragments of MBS may accumulate in P-bodies[5,26–28], although they can be reliably degraded in mammalian cells[11]. We therefore investigated whether the RIDR-induced RNA granules contained MBS fragments alone. We performed two-color smFISH-IF, with probes targeting the open reading frame (ORF) of the mACTB gene (*mACTB-ORF*) and MBS separately. We found that both smFISH signals formed granules colocalizing with the P-bodies after induction, indicating that the mRNAs recruited to P-bodies were not just residual MBS fragments, but also contained the *mACTB-ORF* (Supplementary Fig. 4a, b). To alleviate the degradation artifact in yeast, an MBSv6 sequence was designed to prevent accumulation of MBS-tagged RNA in P-bodies[29]. To further investigate this possible artifact, we applied RIDR to an mCherry reporter containing 24xMBSv6 (*mCherry-MBSv6*) in the 3′UTR driven by the cmv promoter. We co-transfected RIDR and *mCherry-MBSv6* in U-2 OS cells and performed an smFISH-IF experiment 30 min after induction. We observed *mCherry-MBSv6* mRNAs were also recruited into P-bodies (Supplementary Fig. 4c, d). In Fig. 1, we did not observe apparent *mCherry-MBSv5* RNA granule formation in U-2 OS cells. We hypothesized that this might be due to its low expression under the ubc promoter. We repeated the same experiment in U-2 OS cells with *mCherry-MBSv5* driven by a high-expressing cmv promoter and found that *mCherry-MBSv5* mRNAs also formed granules colocalizing with P-bodies 30 min after induction (Supplementary Fig. 4e, f). Taken together, mRNA recruited to P-bodies after induction contained both the ORF and MBS, and is not an artifact of the MS2 system.

Another potential convoluting problem in the RIDR-induced RNA granules is that SMG7C alone may directly recruit MCP or RNA into P-bodies if it is localized there prior to induction. Although it has been reported that full-length SMG7 (SMG7FL) localizes to P-bodies, SMG7C was not observed to do the same[19]. We created fluorescently labeled versions of SMG7C and SMG7FL used in the CID system (FRB-eGFP-SMG7C and FRB-eGFP-SMG7FL) to observe their localization. Similar to previous studies, FRB-eGFP-SMG7C was diffusive in cytoplasm and did not form noticeable granules (Supplementary Fig. 5a), though FRB-eGFP-SMG7FL did weakly colocalize with P-bodies (Supplementary

Fig. 5b). Since we have already confirmed that the granules contained RNA and not just MCP (Fig. 2), we concluded that RIDR-induced RNA recruitment to P-bodies was not simply due to SMG7C tethering, but was a result of SMG7C-induced RNA decay.

## The effect of translation inhibition on induced RNA decay

Previous studies have shown that translation and RNA decay are intimately coupled[30,31]. On one hand, ribosomes may compete with decay machinery for access to mRNA and protect mRNA from degradation[32,33]. On the other hand, it was recently shown that translation may increase mRNA decay rates[34]. To test how translation influences the induced rapid RNA decay, we performed the RIDR experiment in the presence of various translation inhibitors. Cycloheximide (CHX) binds to the ribosomal E-site and inhibits elongation[35]. At high concentrations, it freezes ribosomes on transcripts and disperses P-bodies[6]. Indeed, upon CHX treatment, P-bodies disappeared, and RNA granules no longer formed after induction (Supplementary Fig. 6a, b). The decay of mRNA was also highly reduced (Supplementary Fig. 6c, d). Another translation inhibitor, puromycin, releases nascent peptides and ribosomes from mRNAs. When treated with puromycin, *ACTB-MBS* mRNAs were recruited to P-bodies after induction and the decay was as efficient as the control (Supplementary Fig. 6a–d) This is consistent with the model that loaded ribosomes inhibit rapid decay of mRNA.

## P-bodies provide a kinetic advantage to RNA decay

In the last section, we demonstrated that after induction, *ACTB-MBS* RNAs were rapidly recruited to P-bodies and disappeared -- potentially degraded inside the P-bodies. However, P-bodies are not required for RNA degradation[7,8]. It is unclear what the function is for recruiting RNA to P-bodies during RNA decay. It is plausible that P-bodies offer a kinetic advantage to decay of RNA due to locally concentrated decay factors[7]. Yet, previously it was difficult to visualize mRNA decay in P-bodies due to their stochastic recruitment of RNA and fast decay speed. The synchronous recruitment induced by RIDR amplified the RNA signal in P-bodies, allowing us to quantify the compartment-specific decay kinetics. We developed a kinetic model to describe the RNA trafficking into P-bodies and the subsequent decay dynamics (Fig. 3a, b and Methods). In the most general form, RNAs are recruited into and released from P-bodies with rates $k_R$ and $k_L$, respectively, and the RNA decay rates in P-bodies and the cytoplasm are described by $k_{PB}$ and $k_{CT}$, respectively. To measure these rates, we performed a time-resolved smFISH-IF experiment by fixing cells at different time points after induction. We quantified the number of single mRNAs in the cytoplasm and the integrated intensity of RNA granules in the P-bodies separately (Methods). We normalized the RNA granule intensity into absolute RNA counts by dividing it with the single mRNA intensity (Methods). As a result, we obtained the numbers of mRNAs in the cytoplasm and P-bodies as a function of time. In principle, we could fit the two curves to determine all four rate constants. However, there were not enough features in the curves to unambiguously determine all parameters (Supplementary Fig. 7a). Therefore, we considered three mechanistically interesting special scenarios with only three independent parameters. First, the P-bodies only functioned as a storage site and there was no decay inside ($k_{PB} = 0$). Second, the decay rates in the P-body and cytoplasm were the same ($k_{PB} = k_{CT}$). Third, once mRNAs entered P-bodies, the rate of leaving was negligible ($k_L = 0$). The first two models failed to describe the data (Fig. 3c, d, f). The third model described the data equally well with the full model (Fig. 3e, f, Supplementary Fig. 7a, b). According to the principle of Occam's razor[36], a three-parameter model (Assumption III, $k_L = 0$) is sufficient. Importantly, the fitting revealed that the decay rate of *ACTB-MBS* mRNA in P-bodies was higher than that in the cytoplasm (Fig. 3g). Collectively, this data suggests that RNA decay can occur in P-bodies and that P-bodies offer a kinetic advantage for RNA decay.

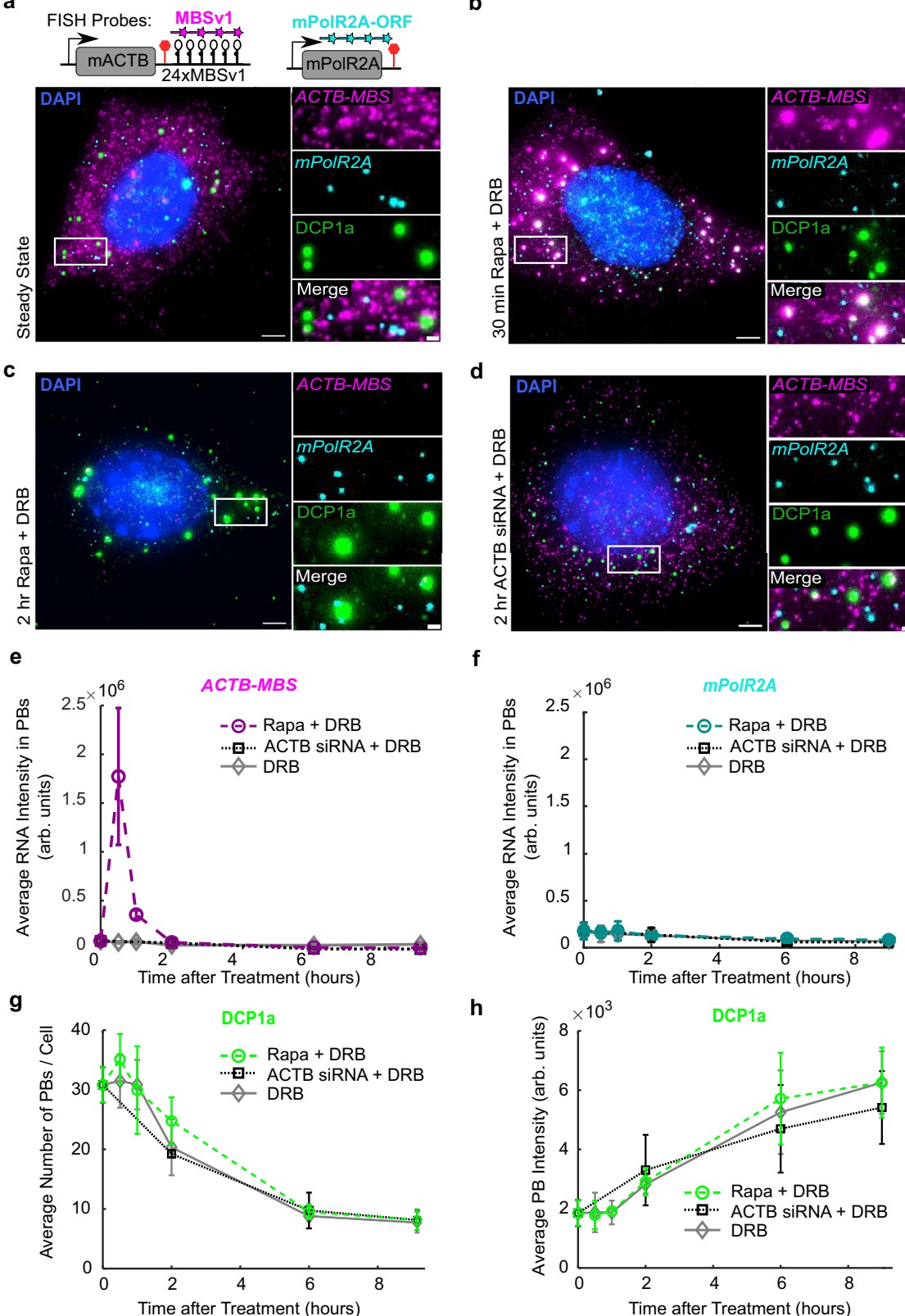

## Fast RNA decay occurs inside the P-bodies

Using mathematical modeling, we showed that RNA decay can occur in P-bodies with a faster rate than in the cytoplasm. To validate this model further, we perturbed the system by depleting key P-body constituent proteins or decay enzymes to observe the difference in induced RNA decay kinetics.

Although we observed RNA granules rapidly formed in P-bodies and quickly dissolved, it was still possible that RNA was only processed there and then released into the cytoplasm for decay afterwards. We demonstrated that this model (Fig. 3c, Assumption I) could not describe the decay kinetics. To convey this directly, we used siRNA to knock down XRN1 (Supplementary Fig. 8a, b), the major 5' ⟶ 3' RNA

**Fig. 2 | *ACTB-MBS* transcripts are recruited to P-bodies after RIDR induction.**
**a–c** ACTB-MBS MEF cells stably expressing RIDR construct were (**a**)
untreated, (**b–c**) induced by Rapa, or (**d**) treated with siRNA against ACTB. DRB was
added to inhibit transcription at time zero. Cells were fixed at different time points
after treatment. smFISH-IF experiments were conducted with FISH probes against
*ACTB-MBS* and *mPolR2A*, and an antibody against DCP1A. Representative images
were shown displaying merged images for FISH and IF channels after (**a**) no treat-
ment; (**b**) 30 min Rapa; (**c**) 2 h Rapa; (**d**) 2 h ACTB siRNA treatments. The white box
was enlarged on the right. *ACTB-MBS* FISH: magenta; *mPolR2A* FISH cyan; DCP1a IF:

green; DAPI: blue. Scale bars: 5 μm for original images, 1 μm for zoomed images.
**e, f** Quantification of integrated intensities of RNA inside P-bodies after induction,
for *ACTB-MBS* (**e**) or *mPolR2A* (**f**) mRNAs. **g, h** Quantification of P-body number per
cell (**g**) and average integrated intensity per P-bodies (**h**). Rapa + DRB: circles; ACTB
siRNA + DRB: squares; DRB alone: diamonds. Error bars represent standard devia-
tion of the means of 3-4 biological replicates. 125-253 cells were quantified per
condition across replicates (the precise number of cells per condition per replicate
are given in the source data).

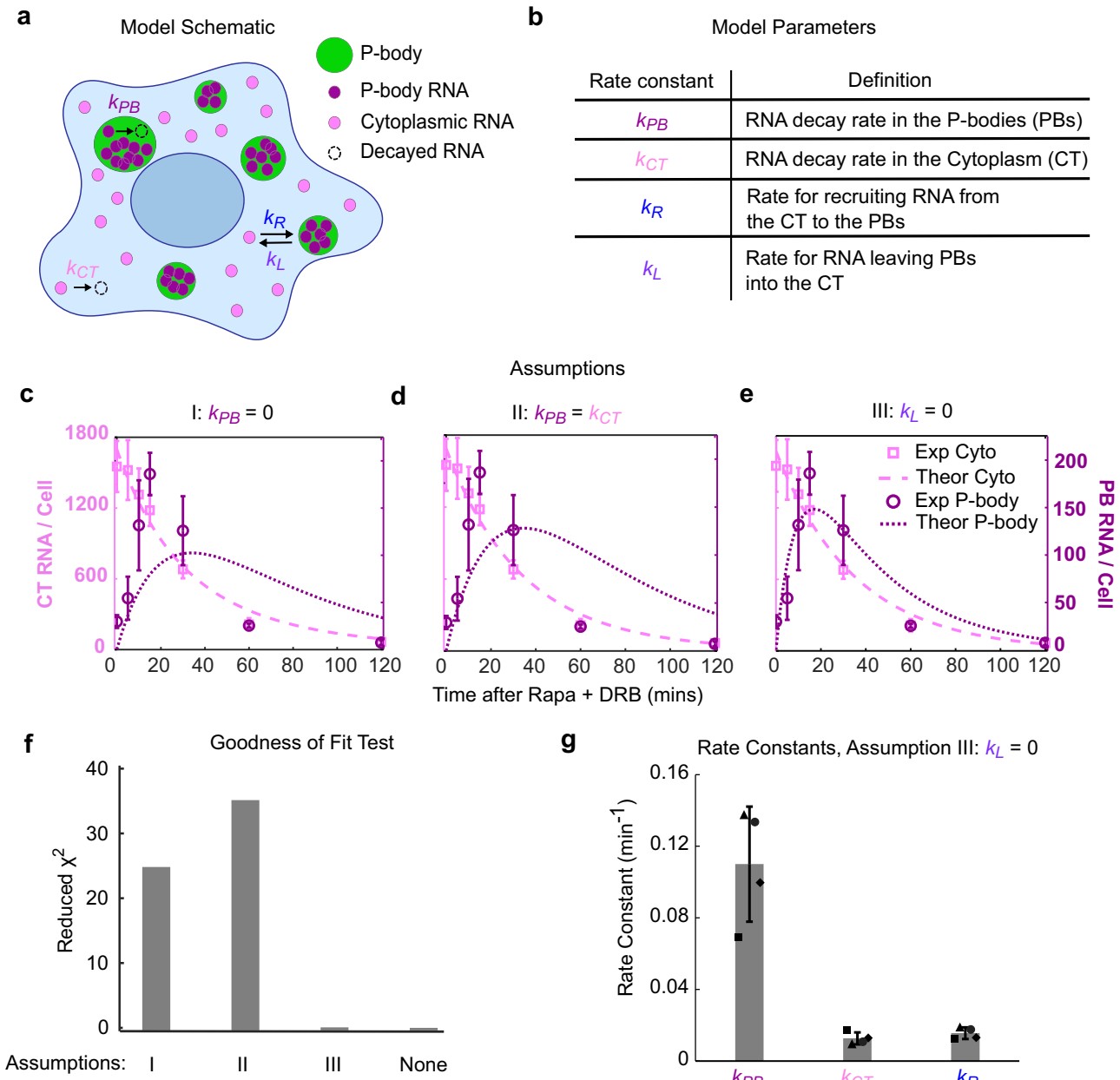

**Fig. 3 | Kinetic Modeling of induced RNA decay in P-bodies and Cytoplasm.**
**a** Schematic of the mathematical model depicting RNA decay, recruitment into and
release from P-bodies. The details are described in main text and Supplementary
Theory. **b** Table with definitions of each kinetic rate constant. **c–e** Fitting results
under different assumptions (**c**) I, no decay in P-bodies:, $k_{PB} = 0$; (**d**) II, decay in
P-bodies and the cytoplasm are the same: $k_{PB} = k_{CT}$; or (**e**) III, RNAs recruited into
P-bodies do not leave: $k_L = 0$. RNA counts in P-bodies: dark magenta; RNA counts in
the cytoplasm: light magenta; Experimental data: symbols; Theoretical fit: lines.

Error bars represent standard deviation of the means of 3–4 biological replicates.
150–445 cells were quantified per condition across replicates (the precise number
of cells per condition per replicate are given in the source data). **f** Reduced $\chi^2$ values
indicating goodness of fit for models in (**c–e**), and with full model (Supplementary
Fig. 5b, c). Lower values indicate better fitting. **g** Model parameters determined
from fitting with Assumption III: $k_L = 0$. Data are presented as mean values of the
fitted parameters across the 3–4 biological replicates. Error bars represent the
standard deviation. Source data are provided as a Source Data file.

exonuclease which is also enriched in P-bodies[37]. RNAi of XRN1 resulted in reduction of the XRN1 protein both in the cytoplasm and P-bodies (Supplementary Fig. 8a), though absolute quantification of the concentration difference in P-bodies versus the cytoplasm using immunofluorescence is challenging. In XRN1-depleted cells, *ACTB-MBS* mRNAs were markedly more enriched in P-bodies after RIDR induction, and the RNA granules persisted much longer compared to treatment with a scrambled non-targeting siRNA control (NC siRNA) (Fig. 4b and Supplementary Fig. 8c). Control mRNAs *mGAPDH* and *mPolR2A* were not recruited nor retained in P-bodies after XRN1 or NC siRNA treatment (Fig. 4c and Supplementary Fig. 8d, respectively). Knocking down XRN1 results in a higher level of cytoplasmic mRNA after RIDR induction compared to the NC siRNA control. However, the accumulation of mRNA in P-bodies is not simply due to increased cytoplasmic mRNA counts. In fact, the percentage of mRNA in the cytoplasm decreased in the XRN1 RNAi condition (Supplementary Fig. 8e). This is consistent with the Assumption III that RIDR-induced mRNAs are recruited into P-bodies and decayed there, as reduced XRN1 levels prolonged the residence time and increased the decaying mRNA levels in the P-bodies.

DDX6, a DEAD box RNA helicase, is essential for P-body formation[38]. When DDX6 was knocked down with siRNA (Supplementary Fig. 8f, g), there were no visible P-bodies (Fig. 4d) and reduced cytoplasmic signals of DDX6 were observed (Supplementary Fig. 8f). As a result, no RNA granules appeared after RIDR treatment, as expected (Fig. 4d). *ACTB-MBS* mRNAs still decay without P-bodies, but

the rate of decay was slower compared to NC siRNA treatment (Fig. 4e). Importantly, the non-targeting mRNAs, *mGAPDH* and *mPolR2A*, decayed with the same rate when DDX6 was knocked down (Fig. 4f and Supplementary Fig. 8i), indicating that the loss of DDX6 itself does not slow down general RNA decay. In sum, this data suggests that P-bodies are required to achieve rapid induced RNA decay.

## RNA decay in P-body is sensitive to stress

In the next set of experiments, we aimed to visualize single mRNAs' recruitment to and decay in P-bodies. There have been controversies about the exact role of P-bodies in RNA metabolism. We observed that mRNAs were recruited to P-bodies and rapidly decayed there. It was puzzling for us that different laboratories have drawn quite contradictory conclusions. During an attempt to capture the RNA dynamics in P-bodies via live-cell imaging, we gained some clues.

We visualized *ACTB-MBS* mRNAs using FKBP-HaloTag-tdMCP[22,39] in live-cell imaging. To label the P-bodies, we stably expressed eGFP-DDX6 at a low concentration in the ACTB-MBS MEF cells. Surprisingly, when we employed the laser power normally used to track single mRNAs, we found that *ACTB-MBS* mRNA was recruited into P-bodies and persisted as long as two hours after Rapa induction (Fig. 5a and Supplementary Movie 2). To verify that the microscope stage-top incubation environment did not perturb the system, we used minimal excitation required for visualizing RNA granules. Indeed, at this condition, we observed the recruitment and dissolution of RNA granules in the P-bodies within one hour,

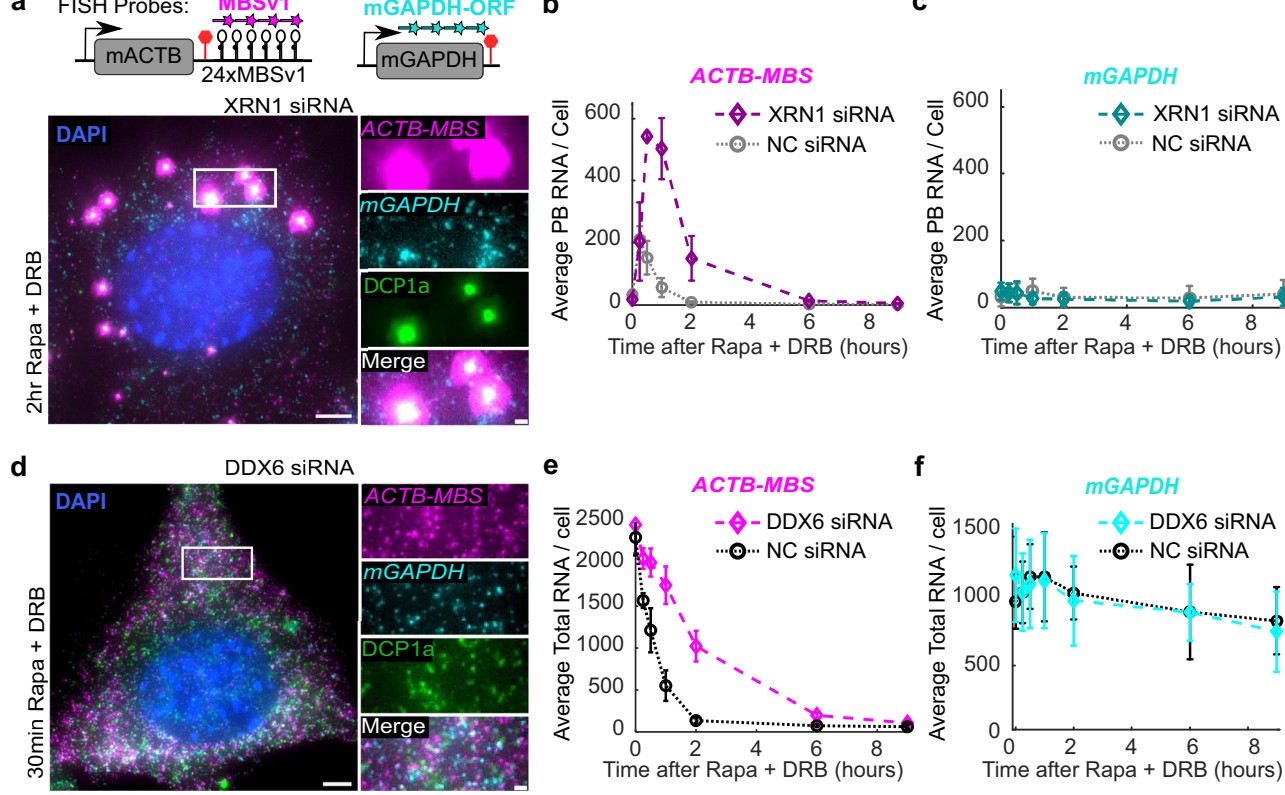

**Fig. 4 | P-bodies are the site of fast RNA decay.** ACT-MBS MEF cells were treated with siRNA against (**a**–**c**) XRN1, (**d**–**f**) DDX6, or (**b**–**c**, **e**–**f**) scrambled siRNA (NC) for 72 h. Cells were treated with Rapa and DRB, and fixed at different time points. smFISH-IF experiments were conducted with FISH probes against *ACTB-MBS* and *mGAPDH*, and antibodies against DCP1a. **a** Representative image at 2 h post induction for XRN1 siRNA treated cells. The white box was enlarged on the right. *ACTB-MBS* FISH: magenta; *mGAPDH* FISH: cyan; DCP1a IF: green; DAPI: blue. Quantification of *ACTB-MBS* (**b**) or *mGAPDH* (**c**) mRNAs in P-bodies over 9 h time course after induction. XRN1 siRNA: diamonds; NC siRNA: circles. **d** Representative image for

DDX6 siRNA treated cell at 30 min post-induction. The white box was enlarged on the right. *ACTB-MBS* FISH: magenta; *mGAPDH* FISH cyan; DCP1a IF: green; DAPI: blue. There are no visible P-bodies under the DDX6 knockdown condition.
**e**, **f** Quantification of total *ACTB-MBS* (**e**) and *mGAPDH* (**f**) mRNA levels over 9 h time course after induction. DDX6 siRNA: diamonds; NC siRNA: circles. Scale bars: 5 µm for original images, 1 µm for zoomed images. Error bars represent standard deviation of the means of 3–4 biological replicates. 193–515 cells were quantified per condition across replicates (the precise number of cells per condition per replicate are given in the source data). Source data are provided as a Source Data file.

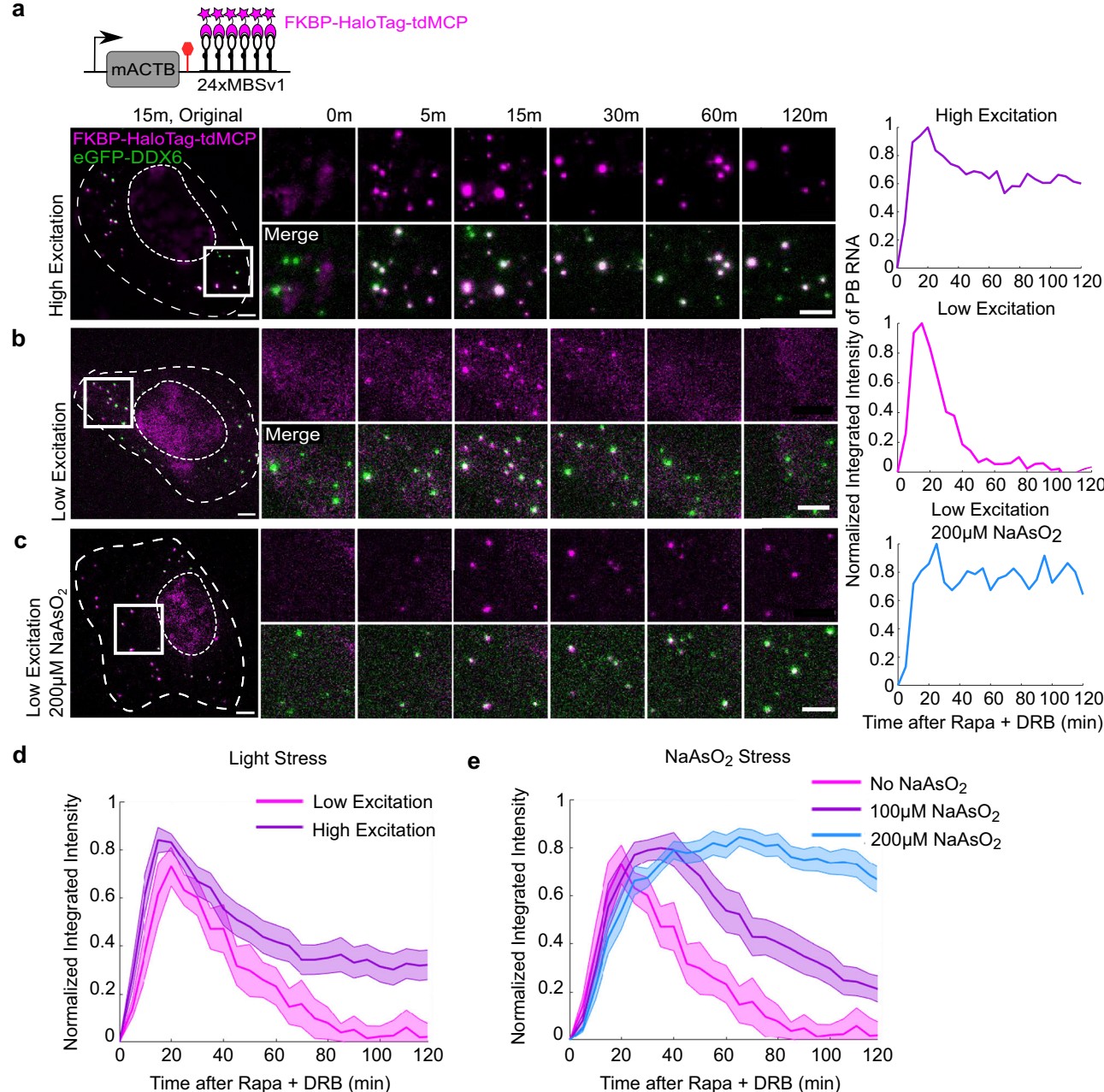

**Fig. 5 | RNA fate in P-bodies is influenced by stress.** Live-cell imaging experiments were performed to track P-bodies (eGFP-DDX6, green) and *ACTB-MBS* (FKBP-HaloTag-tdMCP, magenta) after induction. **a–c** Representative movie montages under different conditions: **a** High excitation laser power for single mRNA tracking was used (Supplementary Movie 2); **b**) minimal laser power sufficient to observe RNA granules colocalized with P-bodies (Supplementary Movie 3); (**c**) minimal laser power and cells were pretreated with 200 µM NaAsO₂ for 30 min (Supplementary Movie 4). The normalized intensity traces of RNA granule intensity of movies (**a–c**) were shown on the right. Scale bars: 5 µm for both original and zoomed images.

**d** Average intensities of RNA granules colocalized with P-bodies under low (magenta) and high (purple) excitation power (number of cells: 8, 14; number of independent experiments: 2, 3; respectively). **e** Average intensities of RNA granules colocalized with P-bodies under low excitation when cells were pretreated for 30 min with 0 µM (magenta), 100 µM (purple), and 200 µM (blue) NaAsO₂ (number of cells: 8, 10, 12; number of independent experiments: 2, 2, 2; respectively). All intensity traces were normalized in each cell before averaging. Data are presented as mean values across each cell. Shaded error bars represent standard error. Source data are provided as a Source Data file.

just like the fixed-cell experiments where no laser excitation was used (Fig. 5b and Supplementary Movie 3). The live-cell RNA granule intensities also matched well with the fixed smFISH measurement after adjusting the amplitude (Supplementary Fig. 9a), indicating that the decay dynamics of *ACTB-MBS* RNA in P-bodies were captured at a low laser power condition.

We hypothesized that the high laser power used for single mRNA imaging created reactive oxygen species and subjected the cells to oxidative stress, hence perturbing the function of P-bodies

and shifting the RNA decay kinetics. To verify this hypothesis, we applied artificial oxidative stress to the cells by pre-treating the samples with varying concentrations of sodium arsenite (NaAsO₂) for 30 min. We imaged the pre-stressed cells after induction at low excitation conditions (Fig. 5c and Supplementary Movie 4). With increasing concentration of NaAsO₂, mRNAs recruited to P-bodies persisted longer (Fig. 5c, e). This suggests that RNA decay dynamics inside P-bodies are tunable, and the function of P-bodies is context-dependent.

It has been reported that stress induced by NaAsO$_2$ can lead to interactions between SGs and P-bodies in the form of docking[40,41], which we also confirmed (Supplementary Fig. 3c). We investigated whether NaAsO$_2$ treatment alone caused recruitment of *ACTB-MBS* RNA signal in the P-bodies. We visualized *ACTB-MBS* mRNAs and P-bodies in the presence of NaAsO$_2$ alone, and did not observe accumulation of RNA signal in P-bodies over 2 h (Supplementary Fig. 9b, c and Supplementary Movie 5). To investigate how induction and stress influence decay of endogenous transcripts, we applied smFISH-IF to measure the *mGAPDH* and *mPolR2A* mRNAs during light or oxidative stress. We used a commercial 24-well blue LED light stimulator (Live Cell Instrument) to mimic the light stress in live-cell imaging experiments. The light stress dosage can be controlled by varying the light intensity and exposure time. To introduce oxidative stress, we treated cells with 200 μM NaAsO$_2$. The endogenous *mGAPDH* and *mPolR2A* transcripts did not enrich in P-bodies during stress, regardless of the presence or absence of Rapa induction (Supplementary Fig. 9d). Decay of these endogenous mRNAs was also not significantly influenced by stress or Rapa induction (Supplementary Fig. 9e).

## Discussion

In this study, we developed a genetically encoded inducible RNA decay system that is fast, specific, and modular. We demonstrated its utility for decaying exogenous as well as endogenous transcripts. RIDR is notably faster than siRNA, reducing the RNA's half-life from 2 to 3 h using siRNA to ~30 min using RIDR. After RIDR induction, we observed that endogenous *ACTB-MBS* mRNAs were recruited to P-bodies. The RNAs recruited to P-bodies are not just MBS fragments, but decaying mRNAs. We examined the compartmentalized RNA decay with mathematical modeling and genetic perturbation. We concluded that RNA decay occurs inside P-bodies and that P-bodies confer a faster decay rate. Surprisingly, we found that the functional role of P-bodies is modulated by cellular stress. Coupling fluorescence imaging with the synchronous induction of RNA decay, RIDR enables investigation of spatiotemporal RNA decay dynamics previously unattainable. The rapid and synchronous decay is important, as it amplifies the RNA signals recruited into P-bodies, allowing us to determine the distinct compartment-specific RNA decay kinetics.

By measuring the recruitment and disappearance of RNA in P-bodies, this study revealed the functional roles of P-bodies. First, we showed that RNA can be degraded in P-bodies. The alternative model that there is no decay in P-bodies does not fit the experimental data (Fig. 3c, Assumption I). Moreover, when the 5′ ⟶ 3′ exonuclease XRN1 was knocked down, the mRNA enriched in P-bodies increased nearly threefold and required more time to disappear, further supporting the model. These findings challenge the notion that P-bodies are only sites for RNA storage. Second, this study suggests that P-bodies provide a kinetic advantage for rapid RNA decay. The measured rate of decay in P-bodies was faster than in the cytoplasm (Fig. 3g). Of note is that these decay rates are likely faster than physiological rates, as the upstream processes have been bypassed during induction. When P-bodies were disrupted by knocking down the essential constituent protein DDX6, no RNA granules appeared, and the rate of total RNA decay was substantially reduced. While DDX6 itself may influence the RNA decay, we showed that it was not the case because the control *mGAPDH* and *mPolR2A* mRNAs were not influenced by DDX6 knockdown. Rapid decay is required for certain transcripts which regulate the cell cycle, apoptosis, and embryonic development[42,43]. It would be interesting to investigate whether P-bodies play a role in regulating the decay of these transcripts. Third, we showed that P-body function was tunable and modulated by cellular stress. This tunability may underlie the variability in the literature regarding the role of P-bodies, as we explain next. There have been controversies about the exact role of P-bodies in RNA metabolism. P-bodies were originally proposed as the location of RNA decay because of the enrichment of RNA decay factors[37]. Later, it

was found that RNA decay does not require the presence of visible P-bodies[7,8] and P-bodies may have dual roles of RNA storage and decay[44]. Even further, it was proposed that P-bodies function mostly as a place for storing mRNA and that there is no decay in P-bodies at all[45]. In previous single molecule studies, mRNAs were observed entering the P-bodies, then remaining there or leaving without decaying[11,46]. To obtain a high signal to noise ratio necessary for visualization of single RNAs, one must typically use high intensity imaging conditions. In this study, the RNA recruitment to P-bodies was synchronized, and therefore amplified the signal in the P-bodies. This amplified signal of the RNA enabled the use of a mild, low intensity imaging condition, which revealed a different phenotype than when we employed higher intensity imaging conditions in an attempt to capture single RNA dynamics. The ability to tune the RNA decay dynamics in the P-bodies under varying levels of stress highlights the context-dependence of RNA fate in P-bodies[47].

The rapid and synchronous RNA decay of highly expressed endogenous transcripts rendered an amplified effect of RNA recruitment to P-bodies upon RNA decay induction. Compared to RNAi, RIDR is much faster; the same observations and kinetic modeling could not be extracted with siRNA treatment. Though we did occasionally observe RNA colocalized with P-bodies after siRNA treatment, RNA didn't accumulate to the degree observed with RIDR. This could be due to the slower kinetics of RNAi compared to RIDR, or perhaps the pathway for RNAi does not result in RNA being recruited to P-bodies. More research must be done to determine the factors that determine which RNAs are brought to P-bodies, and whether they will be stored or decayed upon localization. Though we observed SMG7C tethering caused mRNAs to enter P-bodies for decay, it is possible that the fate of decay in P-bodies is also dependent on the context of the SMG7C tethering.

There are some limitations in the current implementation of RIDR. First, we used the MBS/tdMCP system to tether one CID component to RNA. It is not yet known whether tethering of SMG7C can cause decay of all mRNAs, or just MS2-tagged RNAs. Furthermore, though CRISPR technology has revolutionized gene editing[48], knocking in long tags is still cumbersome. A future improvement would be to target unmodified endogenous RNA with programmable RNA binding proteins like rCas9[49] or catalytically dead CRISPR-Cas13[50–52]. However, we have shown that multiple binding sites are required to achieve efficient knock down with SMG7C. Therefore, signal amplification would be required to target nonrepetitive mRNAs[53,54]. Second, the FRB/FKBP CID system requires rapamycin, an inhibitor for mTor signaling that regulates mRNA translation. We have used low rapamycin concentrations such that translation was not obviously affected (Fig. 1c), so the rapid response of RNA decay should be independent of the mTOR signaling. In the future, other inducible dimerization systems, such as Giberellin[55] or light-induced dimerizers[56,57] can be used to overcome this limitation.

There are previous efforts to manipulate mRNA metabolism in an inducible manner. While sequestering RNA in artificial clusters can offer translational control, it cannot be used to study the RNA's metabolism in physiological contexts[58]. Uncaging methods are another way to inducibly influence RNA functions[59,60]. RIDR provides similar speed and robust tethering efficiency as optogenetic methods developed by Liu and colleagues[61]. The RIDR platform described here is modular, with individual components readily swappable. One can tether other decay factors or RNA regulation factors inducibly to control the mRNA metabolism on demand. By exerting precise spatial and temporal control, the inducible RNA tethering strategy can be used to probe the elusive transient processes that are difficult to study.

## Methods

### Ethical statement

Our research complies with all relevant ethical regulations. No animal or human samples were used in this study.

## Materials availability

Reagents and materials produced in this study are available from Bin Wu pending a completed Materials Transfer Agreement. Constructs will be made available on Addgene.

## Plasmid construction

The mCherry-24xMBSv5 plasmid was described by Wang and colleagues[62]. To clone the mCherry-nxMBSv5 reporters (where $n$ = 0, 1, 3, 6, 12), we used restriction digestion to remove mCherry-24xMBSv5 from the phage-ubc backbone, then used PCR to amplify mCherry plus the number of MBSv5 desired and ligated the mCherry plus nxMBSv5 stem loops into the original backbone. Because MBSv5 is non-repetitive, the PCR of the stem loops was possible.

For direct tethering of RNA decay factor, we cloned SMG7C-Halo-tdMCP-NLS-HA and SMG6PIN-Halo-tdMCP using Gibson assembly. For the inducible RNA decay factor tethering assays focused on SMG7C, we constructed FRB-SMG7C-IRES-FKBP-Halotag-tdMCP-NLS-HA (RIDR) via 4-part Gibson assembly to clone the RIDR construct into a phage lentiviral backbone with a ubc promoter. An FRB-SMG7C Geneblock was ordered to simplify the cloning. To create FRB-BFP-IRES-FKBP-Halotag-tdMCP-NLS-HA (-SMG7C negative control), SMG7C was replaced by BFP using restriction digestion cloning. Prior to stable cell line integration into U-2 OS or ACTB-MBS MEF cells, the RIDR construct was extracted by PCR and cloned into a Tet-On 3 G backbone for Dox-inducibility. The Tet-On 3 G backbone was a gift from Sergi Regot's lab. The inducible expression of RIDR prevents the gene from being lost during long term cell culture.

For live-cell imaging of P-bodies, the phage–UbiC-tagRFP-DDX6 plasmid was ordered from AddGene (#119947) and tagRFP was replaced by eGFP prior to stable integration into the ACTB-MBS MEF cell lines.

## Stable cell line generation

Lentiviral particles were generated by transfecting low-passage HEK293T cells with either FRB-SMG7C-IRES-FKBP-Halo-tdMCP, mCherry-24xMBSv5, or eGFP-DDX6 plasmids along with Generation II viral packaging accessory plasmids. Plasmid transfections were performed using polyethyleneimine (PEI). 48 h following transfection, the viral supernatant was collected, spun down to remove cellular contents, and filtered through a 0.45 μm PVDF filter (Millipore SLHV013SL). The filtered supernatant was applied directly to U-2 OS cells (American Type Culture Collection HTB-96). Viral transduction was performed sequentially by first infecting U-2 OS cells with mCherry-24xMBSv5 and performing fluorescence activated cell sorting (FACS) for mCherry positive cells. This positive population was then infected in the same manner with FRB-SMG7C-IRES-FKBP-Halo-tdMCP and sorted for high–expression cells.

Immortalized MEF cells with 24x MBS at the endogenous ACTB locus (ACTB-MBS MEF) were a gift from Robert Singer's lab. The ACTB-MBS MEF cells were stably integrated with the dox-inducible expression of FRB-SMG7C-IRES-FKBP-Halo-tdMCP in the Tet-On 3 G system and sorted for HaloTag expression using flow cytometry. ACTB-MBS MEF cells used for live-cell imaging were infected in the same manner as above with DDX6-eGFP plasmid for stable integration, and then sorted for GFP expression using flow cytometry.

## Cell culture and transfection

U-2 OS (American Type Culture Collection HTB-96), HEK293T (American Type Culture Collection CRL-1573), and ACTB-MBS MEF cells were grown in DMEM (Corning, 10-013-CV) supplemented with 10% (v/v) FBS (Millipore Sigma, F4135-500ML), 100 U/ml penicillin, and 100 μg/ml streptomycin (Millipore Sigma, P0781) and maintained at 37 °C and 5% $CO_2$. Cells were passaged every 2–3 days once they reached ~75% confluency. Cells were tested monthly for mycoplasma infection and were always negative.

XtremeGeneHP was used to transfect plasmids into HEK293T cells for use in flow cytometry. For 24-well dishes, each well received 250 ng total plasmid DNA. For flow cytometry experiments, 50 ng of plasmid DNA for the reporter mRNA and 200 ng plasmid DNA containing the RNA decay factor were mixed with serum-free DMEM to a final volume of in 25 μL. For each well of a 24-well dish, 1 μL XtremeGeneHP was combined with 24 μL of serum-free DMEM and incubated in a MasterMix for 5 min at room temperature. After incubation, 25 μL of plasmid DNA mix and 25 μL of incubated XtremeGeneHP mixture were combined by gentle pipetting, then incubated together for 15 min at room temperature. After the second incubation, 50 μL XtremeGeneHP and plasmid DNA mixture was added dropwise to each corresponding well. Immediately following transfection, Rapa or DMSO control were added to the cells. Cells were transfected overnight, then prepared for flow cytometry the next morning.

For RIDR vs RNAi benchmarking experiments (Figs. 1–2) cells were prepared in the manner described in the RIDR kinetics section. Lipofectamine® RNAiMAX (Thermo Fisher) was used to transfect siRNAs. ON-TARGET pooled siRNAs of ACTB (IDT mm.Ri.Actb.13.1-3), custom designed mCherry siRNAs or OFF-TARGET IDT negative DsiRNAs (NC) were used. For each well of a 24-well dish, a mix of 5 pmol (1 μL of 5 μM pooled siRNA) + 24 μL OptiMem was combined with a mixture of 1.5 μL RNAiMAX + 23.5 μL OptiMem for a total of 50 μL per well. The siRNA / OptiMem / RNAiMAX mixture was mixed gently, incubated for 5 min at room temperature, then added dropwise into each well of 24-well. DRB was also added at the time of siRNA treatment, at a final concentration of 100 μM. DRB was also added at the time of siRNA treatment, at a final concentration of 100 μM.

For DDX6 and XRN1 knockdown experiments (Fig. 4) Lipofectamine® RNAiMAX was also used to transfect siRNAs. ON-TARGET pooled siRNAs of DDX6 (IDT mm.Ri.Ddx6.13.1-3), XRN1 (IDT mm.Ri.Xrn1.13.1-3) and OFF-TARGET IDT negative DsiRNAs were used. On the evening of Day 1, 100,000 ACTB-MBS MEF cells were plated per well of a 6 well plate. Cells were incubated overnight. On the morning of Day 2, for each well of a 6-well dish, the first RNAiMAX transfection was performed with 7.5 μL RNAiMAX + 67.5 μL OptiMem combined and added to a mixture of 25 pmol siRNA (5 μL of 5 μM pooled siRNA) + 70 μL OptiMem, for a total of 150 μL. The siRNA + OptiMem was mixed gently, incubated for 5 mins at room temp, then added dropwise into the 6-well dish. The cells were incubated for 24 h. On the morning of Day 3, 15,000 cells were replated onto 12 mm fibronectin-coated coverslips (Electron Microscopy Science, 72290-03) in a 24 well. At the end of Day 3, the second transfection was performed. For each well of 24-well dish, a mix of 5 pmol (1 μL of 5 μM pooled siRNA) + 24 μL OptiMem was combined with a mixture of 1.5 μL RNAiMAX + 23.5 μL OptiMem for a total of 50 μL per well. The siRNA and OptiMem were mixed gently, incubated for 5 min at room temperature, then added dropwise into each well of 24-well with coverslips. At the end of Day 4, the medium was replaced. To induce expression of RIDR construct, 1 μg/mL Doxycycline was added to the medium and incubated overnight. On the morning of Day 5, ~72 h after initial transfection, the cells were ready for RIDR treatment. siRNA sequences can be found in Supplementary Data 1.

## Flow cytometry

HEK293T cells were plated onto a 24-well with 25,000 cells per well, incubated for 24 h. The cells were transiently transfected with 250 ng total plasmid using the transfection reagent XtremeGeneHP according to the manufacture's instruction. 100 nM rapamycin or DMSO were applied to the cell immediately after transfection. After 14–16 h, the cells were labelled with 10 nM JF503-Halo-Ligand for 1 h, washed for 30 min, trypsinized, resuspended into complete DMEM, then filtered through a cell strainer (Corning 352235). Flow cytometry data was collected on a Thermo Attune NxT flow cytometer. To calculate knockdown efficiency, the JF503-positive cells were gated

(Supplementary Fig. 1a) and the geometric mean of fluorescence intensities for mCherry and JF503 channels were calculated in the FlowJo software individually, and compared to the respective control conditions (-Rapa, -SMG7C).

### RIDR time course experiments

For RIDR time course experiments involving U-2 OS cells expressing mCherry-24xMBSv5 (Fig. 1), 50,000 cells were plated the night before on 12 mm coverslips (Electron Microscopy Service, 72290-03). For fixed-cell RIDR time course experiments involving ACTB-MBS MEF cells (Fig. 2), 25,000 cells were plated on fibronectin-coated 12 mm coverslips the evening before a time course experiment. Plating procedures for RIDR kinetics experiments after siRNA treatment (Fig. 4) or live-cell imaging (Fig. 5) are explained in the Cell Culture and Transfection section.

For all time course experiments (Figs. 1–5) cells were treated overnight with $1\,\mu g/mL$ Doxycycline to induce expression of the RIDR construct. The next morning, cells were labelled with JF646 Halo-ligand at the start of the time course experiment prior to the addition of Rapa or DRB Rapamycin powder (LC Laboratories, R-5000-100MG) was dissolved in DMSO for a final stock concentration of 10 mM. Rapa aliquots were stored at $-20\,°C$ for long term storage. Prior to a RIDR experiment, a fresh Rapa aliquot would be used and diluted further to $100\,\mu M$ in DMSO. Transcription inhibitor 5,6-dichloro-1-beta-D-ribofuranosylbenzimidazole (DRB) powder (Millipore Sigma, D1916-50MG) was dissolved in DMSO for a final stock concentration of 100 mM. DRB aliquots were stored at $-20\,°C$ for long term storage and fresh aliquots were used for each experiment.

Both Rapa and DRB were further diluted 1000x to reach their final concentration in the cell culture medium. To ensure adequate dispersion of the drug(s), the appropriate amount of Rapa and/or DRB was added to a fresh microcentrifuge tube, $\sim 200\,\mu L$ of media was removed from the well or dish where the drugs were to be added, the drug(s) were resuspended in this medium by pipetting, then added back into the well or dish drop-wise. Final concentrations in the wells or dishes for Rapa and DRB were 100 nM and $100\,\mu M$, respectively. Steady State condition was not treated with Rapa or DRB. Cells were kept in a stage top incubator (Tokai Hit) maintained at $37\,°C$ with 5% $CO_2$ and protected from light.

### Fluorescent in situ hybridization

The RNA single-molecule FISH (smFISH) using 20mer DNA oligo probes was adapted from the work of Raj and colleagues[63] and described in detail by Gaspar and colleagues[64]. In brief, DNA oligos were ordered from Integrated DNA Technology and labeled in house with Cy3, Atto590, or Cy5. ACTB-MBS MEF cells were seeded on 12 mm glass coverslips (Electron Microscopy Service, 72290-03) that were coated for 30 min with 1:400 dilution of fibronectin, (Sigma-Aldrich F1141-2MG) in DPBS and cultured overnight. After fixation with 4% paraformaldehyde and permeabilization with 0.1% of Triton, cells were incubated with 20–40 nM probes in hybridization buffer for 3 h at $37\,°C$. The unbound probes were washed away with 10% formamide and the coverslips were mounted on microscope slide using ProLong Diamond Antifade Mountant containing DAPI (Thermo Fisher Scientific, P36962) for nuclear staining.

FISH probes targeted the MBSv5/MBSv6 or mACTB-ORF/MBSv1 region for U-2 OS cells or ACTB-MBS MEF cells, respectively. Internal controls *hPolR2A*, *mGAPDH* and *mPolR2A* FISH probes targeted the ORF of each gene. Coverslips were mounted onto microscope slides with Prolong Diamond overnight and sealed with clear nail polish after curing the next day. The RNA FISH probe sequences are listed in Supplementary Data 2.

For smFISH combined with immunofluorescence, 1:1000 dilution of rabbit anti-DCP1a (Abcam ab183709), 1:100 rabbit anti-XRN1 (Bethyl Laboratories A300-443A-M), 1:1000 rabbit anti-DDX6 (Bethyl Laboratories A300-461A), or 1:100 rabbit anti-G3BP (Aviva Systems Biology ARP37713_T100) were used as primary antibodies. A 1:5000 dilution of goat-anti-rabbit IgG (H + L) Alexa Fluor 750, Invitrogen A-21039, was used as the secondary antibody for all primary rabbit-derived antibodies.

### Fluorescence microscopy

The fixed samples were imaged on an automated inverted Nikon Ti-2 wide-field microscope equipped with 60x, 1.4NA oil immersion objective lens (Nikon), Spectra X L.E.D. light engine (Lumencor), and Orca 4.0 v2 scMOS camera (Hamamatsu). The live-cell experiments were performed on a custom microscope built around Nikon Ti-E stand. The excitation was through HTIRF (Nikon) with an LU-n4 four laser unit (Nikon) with solid state lasers with wavelengths 405, 488, 561, and 640 nm. The main dichroic was a quad band dichroic mirror (Chroma, ET-405/488/561/640 nm laser quad band set for TIRF applications). The imaging was done through the 100x 1.49NA oil immersion objective (Nikon). To achieve simultaneous 2-color imaging, we used a TriCam light splitter into three separate EMCCD cameras (Andor iXon Ultra 897) with ultraflat 2 mm thick imaging splitting dichroic mirrors (T565LPXR-UF2, T640LPXR-UF2). A band pass emission filter was placed in front of each camera, respectively (ET525/50 m, ET595/50 m, and ET655lp). The microscope was also equipped with an automated XY-stage with extra fine lead-screw pitch of 0.635 mm and 10 nm linear encoder resolution and a Piezo-Z stage (Applied Scientific Instrumentation) for fast Z-acquisition. A microscope stage top incubator (Tokai Hit, Model) is used to keep the sample at $37\,°C$, 5% CO2 and saturating humidity. The whole microscope was under the control of Nikon Elements v4.8 for automation.

### Live-cell Imaging

100,000 MEF cells stably expressing tet3G-FRB-SMG7C-IRES-FKBP-Halo-tdMCP and DDX6-eGFP were plated on a 35 mm with 20 mm micro well #1.5 cover glass bottom (Cellvis, D35-20-1.5-N). Cells were treated with $1\,\mu g/mL$ Doxycycline overnight. The next morning, cells were incubated with 100 nM JF646 Halo Ligand[65] for 30 min, then rinsed once in complete DMEM with 10% FBS and 1% PenStrep, and transferred to a $37\,°C$ 5% $CO_2$ incubator for at least 30 more minutes to equilibrate prior to imaging. During live-cell imaging, the cells were kept at $37\,°C$ with humidity control on a Tokai Hit stage top incubator. For data collection, each cell was imaged every 5 min for 2 h with 100 ms exposure time. Low excitation conditions involved excitation with 2% 488 and 2% 640 laser stimulation, while the high excitation condition involved excitation with 2% 488 and 10% 640, stimulating light stress. Cells imaged after Sodium Arsenite stress were pre-treated with either $100\,\mu M$ or $200\,\mu M$ Sodium Arsenite 30 min prior to imaging in low excitation conditions. Analysis of live-cell imaging was done using u-track v2[66]. P-bodies were detected in the eGFP-DDX6 channel using the Point Source Detection algorithm. The intensity of the FKBP-HaloTag-tdMCP channel was measured over time in the regions segmented by the detected P-bodies, and the intensity was summed at each time point. To produce the Supplementary Movies, max projected images were background subtracted in each channel using the rolling ball algorithm in ImageJ.

### Light and Arsenite stress quantification by smFISH-IF imaging

25,000 ACTB-MBS MEF expressing tet3G-RIDR cells were plated on fibronectin coverslips and grown overnight. Dox was added at the time of plating. 12–16 h later, cells were labeled with 10 nM JF503 Halo Ligand for 2.5 h prior to fixation. Directly after labelling, cells were pre-treated with 30 min of light or arsenite stress. Cells treated with light stress were subjected to light illumination using a Live Cell Instruments (LCI) 24-well blue-light illuminator. The LCI blue-light stimulator was programmed to illuminate the cells constantly at 50% intensity. Arsenite-stressed cells were treated with $200\,\mu M$ NaAsO$_2$. After 30 min

of pre-stress treatment for both light and arsenite stress, Rapa and DRB were added to the cells at a final concentration of 100 nM Rapa and 100 μM DRB, alongside unstressed controls. smFISH-IF and imaging were conducted as previously described.

### Image analysis and quantification of RNA in smFISH experiments

We used an in-house RNA detection platform called uLocalize[67] to count the compartmentalized P-body and cytoplasmic mRNAs separately. All custom code for smFISH-IF analysis and theoretical modeling can be found at (https://doi.org/10.5281/zenodo.7922686) and is summarized here:

To detect P-bodies in the DCP1a IF channel, we filter the image with Laplacian of Gaussian filter and segmented the area using an intensity threshold. Single RNAs were detected using a Local Maximum detection algorithm, and the single RNA intensity was determined by fitting the spot to a 3D Gaussian function to extract the center and the amplitude. To quantify RNAs in P-bodies, we measured the integrated intensity of the max projected RNA channel in the segmented P-body area. The integrated intensities of the RNA granules were normalized to RNA counts by dividing the median max projected single RNA intensity. Finally, RNA counts of all P-bodies in single cells were summed to obtain the total P-body RNA counts used to fit the mathematical models. Cytoplasmic RNA was counted as all the detected RNA in the cytoplasm excluded from the segmented P-bodies. These RNA counts in cytoplasm and P-bodies were summed to obtain the total RNAs in single cells.

### SMG7C vs SMG7FL P-body colocalization measurements

U-2 OS cells were plated on fibronectin coated type I German glass coverslips. 24 h after plating, cells were transfected with either FRB-eGFP-SMG7 (full-length SMG7) or FRB-eGFP-SMG7C (C-terminus of SMG7) using Xtreme gene HP per the manufacturer's instructions. An equal mass of FRB-SMG7 and FKBP-MCP constructs was delivered. The following day, cells were incubated for 30 min with final concentration of 10 nM JFX549 HaloTag ligand and 100 nM rapamycin for the indicated samples. Immunofluorescence was performed with a 1:5000 dilution of rabbit anti-DDX6 primary antibody (Bethyl Laboratories, A300-461A) and 1:1000 dilution of Alexa Fluor goat anti-rabbit IgG secondary antibody (Invitrogen, A21039).

### Supplementary Theory: Mathematical model to describe the compartmentalized RNA decay

Cytoplasmic and P-body RNA counts were calculated as described in Methods and fit to a mathematical model.

Model parameters definitions are described in Fig. 3a, b. The differential equations used to describe this simple kinetic model are as follows:

$$\frac{d\text{RNA}_{CT}}{dt} = -k_{CT} \times \text{RNA}_{CT} - k_R \times \text{RNA}_{CT} + k_L \times \text{RNA}_{PB} \quad (1)$$

$$\frac{d\text{RNA}_{PB}}{dt} = -k_{PB} \times \text{RNA}_{PB} + k_R \times \text{RNA}_{CT} - k_L \times \text{RNA}_{PB} \quad (2)$$

The differential equations were solved analytically in Mathematica. The solution was implemented in MATLAB to fit the cytoplasmic and P-body RNA counts simultaneously using the nonlinear Least Squares Fitting algorithm.

### Translation inhibition

25,000 ACTB-MBS MEF cells were plated on fibronectin-coated coverslips. Cells were treated with 1 μg/mL Doxycycline overnight to express the RIDR construct. The next morning, cells were incubated with 10 nM JF503 Halo Ligand for 2 h prior to fixation. Translation

inhibitors were added to the sample 10 min prior to the Rapamycin + DRB or DRB control in each timepoint. Translation inhibitors were added at a final concentration of 100 μg/μL for Puromycin, 100 μg/μL for cycloheximide. smFISH-IF was then performed as described above.

### Western blot

For DDX6 and XRN1 RNAi experiments, 100,000 ACTB-MBS MEF cells were treated as described in the Cell Culture and Transfection section. Cells were harvested by scraping and pelleted at 500 x g for 2 min. Cell pellet was resuspended in 30 μl ice-cold lysis buffer (50 mM HEPES pH 7.4, 150 mM KOAc, 15 mM MgOAc₂, 1% triton, leupeptin, pepstatin, PMSF, 1x EDTA-free Complete (Sigma 11873580001), 2 U Turbo DNase/ml (ThermoFisher AM2238), then gently pipetted 10 times to lyse. After incubating 5 min on ice, lysates were clarified by centrifugation at 20,000 x g for 10 min at 4 °C and the supernatant was transferred to a new tube. Samples were electrophoresed in a 4% SDS-PAGE gradient gel, transferred to a PVDF membrane, and blotted overnight with a 1:1000 dilution of rabbit anti-XRN1 primary antibody (Bethyl Laboratories A300-443A-M), 1:10000 dilution of rabbit anti-DDX6 primary antibody (Bethyl Laboratories A300-461A), or 1:250 rabbit anti-Ribosomal Protein S3 primary antibody (Santa Cruz sc-376008) as a control. Samples were then incubated with 1:5000 mouse anti-rabbit IgG HRP (Santa Cruz sc-2357) secondary antibody for 1 h at room temperature then washed 3 times for 10 min in 1X Tris-Buffered Saline, 0.1% Tween® 20 Detergent (TBST). All incubation steps were done with gentle rocking. Samples were visualized on a Bio-Rad Chemidoc Imager.

### Statistics & reproducibility

No statistical method was used to predetermine sample size. No data were excluded from the analyses. The experiments were not randomized. The investigators were not blinded to allocation during experiments and outcome assessment.

### Reporting summary

Further information on research design is available in the Nature Portfolio Reporting Summary linked to this article.

## Data availability

The raw imaging data supporting the findings of this study are too large to share on a public platform, but will be available from the corresponding authors upon request within 4 weeks. The source data in the figures are provided with this paper. Source data are provided with this paper.

## Code availability

The analysis code that supports the findings of this study is available in GitHub (https://doi.org/10.5281/zenodo.7922686).

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

## Acknowledgements

We thank members of the Wu and Inoue Labs for helpful discussion. We are also thankful to Boyang Hua for assistance with western blots. This work was supported by the National Institutes of Health (Grant # R01GM136897 to B.W., #R35GM149329 to T.I., #R35GM150941 to Y.L.) and Pew Charitable Trust (Award ID 00030601) to B.W. L.A.B. was supported by N.I.H. Training Grant (T32 GM008403). L.W. was supported by N.I.H. Training Grant (T32 GM007445).

## Author contributions

L.A.B. and B.W. designed the experiments. L.A.B. conducted most experiments and analysis. L.W. assisted with revision experiments and analysis. L.A.B. and B.W. wrote the paper. Y.L. and T.I. advised on experimental design and interpreting data. B.W. supervised the project.

## Competing interests

The authors declare no competing interests.
