## [Peer Review File · Nature Communications]

Reviewers' Comments:

Reviewer #1:

Remarks to the Author:

In the manuscript by Blake et al., the authors developed a rapid inducible RNA decay system, RIDR, which employs the MS2 system and the inducible tethering of SMG7C. Using RIDR, the authors showed that the tagged mRNA was targeted to the P-bodies where the mRNAs were degraded more rapidly than in the cytoplasm. RNA turnover is difficult to capture because of its stochastic nature. RIDR overcame this challenge by inducing synchronous RNA decay, thus is a useful tool to study RNA metabolism. The conclusions of this study were supported by well-designed experiments. The manuscript will be further improved when the following questions are addressed.

Major points:

1. In figure 3 and line 540-541, the authors built a quantitative model to estimate the RNA decay rate in P-bodies. The RNA synthesis rate was not taken into account. And "there were not enough features in the curves to unambiguously determine all parameters". To improve the model, the authors could perform additional experiments under the condition of no Rapa. First, the RNA decay rate in cytoplasm, k_{CT} , could be determined by inhibiting transcription and monitoring the RNA level over time without adding Rapa. Next, the RNA synthesis rate could be derived from the RNA steady state level when no Rapa was added. These additional experimental data will help more accurately determine the parameters in the model.

2. How do light stress and oxidative stress affect the decay of GAPDH and PolR2A mRNA?

3. In figure 5, the authors monitored the RNAs in P-bodies by measuring the Halo signal intensities. There are two populations of FKBP-Halo-MCP in the cells, the ones that are associated the MS2-tagged mRNA and the ones that are not. Do the latter population also move to P-bodies through the induced tethering of SMG7C? If so, the authors need to use an alternative approach, such as smFISH, to monitor RNA decay in P-bodies.

Minor point:

Line 36, "specific a" should be "a specific".

Reviewer #2:

Remarks to the Author:

Blake et al., "A Rapid Inducible RNA Decay system reveals fast mRNA decay in P-bodies"

In this manuscript, the authors describe a tool they have engineered to quantitate RNA degradation rates in living cells.

The system consists of two halves which can rapidly be brought together in vivo by addition of rapamycin. The first half consists of an RNA decay factor (SMG7C) fused to the FKBP-Rapamycin binding domain (FRB). The other half of the pair is made out of FKBP fused with the RNA-binding MS2 coat protein MCP), which recognizes MS2 binding sites in RNA. In the presence of rapamycin, FKBP rapidly binds to FRB, bringing the decay factor close to the mRNA and initiating its degradation. In addition, a target mRNA is labeled with MBS sequences where the MCP can bind, and the FKBP/MCP constructs includes a Halo tag to enable detection by fluorescence of the FKBP/MCP protein. This is put downstream of an mCherry tag, and so degradation of the mRNA induced by SMG7C being recruited there can be monitored in parallel by decreasing mCherry fluorescence.

This construct is used to study the decay rate of selected tagged mRNAs. It was found that mRNA

levels could be reduced by 95 % within 2 h, which is considerably more efficient than RNAi. In the course of these observations, it was found that upon induction of RIDR, labeled RNA accumulates at P-bodies. Quantification of RNA degradation within P-bodies and diffusely in the cytoplasm demonstrated that accelerated RNA decay, relative to the cytoplasm, takes place in P-bodies. The alternative possibility would be that labeled RNA migrated to P-bodies simply for storage.

Interestingly, it turns out that the estimated RNA degradation rates in P-bodies differ greatly between standard conditions and stress (allegedly oxidative stress). The signal duration of the labeled mRNA from P-bodies is greatly prolonged when the cells are exposed to high intensity laser, or arsenite. The authors argue that during stress, the physiology of PBs changes from rapid mRNA degradation to some other, yet undefined state. For a long time, there has been contradictory statements in the literature whether P-bodies are sites of RNA decay, as originally held, or whether they are repositories of decay factors and the majority of RNA degradation takes place outside of them in the cytoplasm. Indeed, it is conceivable that these new findings could reconcile these opposing views.

The experimental design in the paper is well conceived, and the results properly analyzed and interpreted. There are innovative and basic science aspects of this work. Although the RIDR system was custom built for this paper and underlie all the findings in it, it uses already established components for chemically inducible dimerization. I find the basic science findings more important here, as they shed light on fundamental issues of how P-bodies are related to RNA decay. This paper should have a significant impact and be of interest to a wide audience in the RNA biology field, and biomedical sciences at large.

The writing style in the manuscript is well balanced and gives enough experimental detail without sacrificing overview. The rationale for designing the test system and performing the experiments are given with the proper background from previous work from other labs. The selection of literature references is adequate, they cite most of the relevant publications in the field.

Overall, this is a substantial contribution that deserves publication in a high-profile journal. There are however some issues that need to be addressed:

General point:

SMG7, the mRNA decay factor used in the constructed system, is itself located in P-bodies (Unterholzener and Izaurralde 2004). Can this affect the localization of degrading mRNAs to P-bodies? The authors do not discuss this issue. In the same publication, it was shown that SMG6 does not localize to P-bodies but to separate granules. The authors of the present manuscript could rule this possibility in or out by testing their SMG6-based construct in parallel experiments as with SMG7.

To be discussed in the manuscript:

The nature of the engineered inducible degradation system presented here is such that the start of degradation is forced by addition of rapamycin, leading to the joining of the two CID components which brings the RNA decay factor SMG7C in vicinity of the mRNA. This means that the physiological regulation of degradation of the mRNA under study is short-circuited, and the events leading up to the degradation itself are eliminated. Could artifacts arise this way? Could degradation rates be overestimated because the decay machinery is running without the normally imposed constraints?

Further, can the addition of many copies of MBS sequences to the 3'-UTR and expression of the MS2 coat protein influence the results? The MCP can form aggregates when expressed at high levels, can they drag the attached mRNAs with them into punctate structures?

Also, the inducible system relies on SMG7C for RNA degradation. SMG7 acts in the NMD pathway, downstream of e.g. Upf1 and SMG5. It was shown previously (Unterholzner and Izaurralde 2004) that SMG7 tethered to a transcript will execute its degradation, even if the premature termination

codon normally triggering NMD is eliminated. The question arises if RNA-tethered SMG7 will degrade any mRNA, irrespective of whether that mRNA is a natural NMD target or not. To mutate a premature termination codon in a natural NMD target and show that SMG7 will still degrade it, does not prove that SMG7 will act on all mRNAs, or even on a large fraction of the mRNA population. The authors should discuss the generality of their system against this background.

Specific points:

Line 214: "decay rate in P-bodies was significantly higher than in the cytoplasm" Does this refer to the half-life of a particular mRNA species, or to the total degradation capacity of P-bodies (i.e. taking into account the total number of RNA molecules of a certain species)?

Line 220: P-body constituent proteins are also present in the cytoplasm, so the result of depleting them will depend on their relative concentration in P-bodies or the cytoplasm.

Line 226: The interpretation of this result depends on Xrn1 being enriched in P-bodies – but have the authors or anybody else shown that it is enriched?

Line 260: Arsenite is a classical strong inducer of stress granules (oxidative stress by other agents is not a strong stress granule inducer). Arsenite has several effects on cells, not necessarily related to oxidative stress. With this in mind, is it possible that the prolonged signal of labeled mRNA in P-bodies is due to an increasing background under these conditions of stress granules, which are sometimes seen adjacent to P-bodies?

Line 281: Knocking down XRN1 leads to accumulation of labeled mRNA in P-bodies, which also persisted longer. But eliminating XRN1 will affect mRNA degradation globally, also in the cytoplasm. Maybe the labeled mRNA accumulates simply because its concentration in the cytoplasm also increases, and it has nowhere to go to be degraded?

Minor comments:

Line 75: "disappearance of signal being frequently confounded by imaging artifacts" Could the authors clarify what they mean here?

Reviewer #3:

Remarks to the Author:

A Rapid Inducible RNA Decay system reveals fast mRNA decay in P-bodies
by Blake et al.

Blake et al. present a rapid inducible system that allows to target a specific RNA of interest for RNA decay in living cells. The system builds on the recruitment of the C terminus of SMG7 via the MS2 coat protein (MCP) and the integration of MS2 stem loops (MBS) into the RNA of interest. Inducibility is achieved via the chemically inducible dimerization system which brings together SMG7C and MCP upon addition of rapamycin. The authors perform a series of experiments to establish the system and validate its performance for both ectopically and endogenously expressed MS2-tagged RNAs of interest. Of note, they observe an accumulation of the tagged mRNAs in P-bodies. Following up on this observation, they test the impact of translation and different RNA decay factors. They also present a mathematical model to support that the trapping of RNAs in P-bodies allows for faster mRNA decay.

Overall, the authors introduce an elegant tool for the specific targeting and degradation of RNA molecules. They demonstrate that the system works very fast and efficient and allows for the observation of mRNA decay kinetics at the level of individual RNA molecules. Since single-molecule observations will be key to ultimately unravel RNA regulation in cells, I think that the developed tool will be a valuable asset for the community. Regarding the second part of the manuscript, my major concern is whether the observed P-body localisation could be influenced by the MS2 loops used to tag the RNAs of interest (see below).

Major points:

1. As the authors point out, rapamycin itself has an influence on translation which may in turn impact on mRNA decay. Can the authors quantitatively measure this effect? Could this relate to the ~10% Halo KD observed across conditions in Figure 1c? In addition, the minimal rapamycin concentration needed for efficient dimerisation should be established in titration curves.

In this context, Figure 1c is central to demonstrating the functionality of the RIDR system and the influence of its components. Yet, the information provided is not sufficient to evaluate the obtained results. It should be explained more clearly how the data were normalised and how the "% KD efficiency" values were obtained. Unnormalised data and FACS measurements for individual samples (as in Figure 1a) could be added in the Supplementary to allow for a full evaluation of the data.

3. The authors use MBSV5 as MS2 loops for tagging the RNA of interest. However, multiple studies in yeast suggested that MS2 loops can interfere with 5'-to-3' degradation and produce RNA fragments that end up in P-bodies (PMID: 26092944, 12730603, 28096443, 27090788). This has also been explicitly shown for the MBSV5 loops that were used in the present study (PMID: 29131164). If this would also be the case in the system presented here, it would have strong confounding effects on the observations made.

Further investigations regarding this potential artifact are necessary to ensure the validity of the conclusions. To assess whether complete RNAs or decay fragments are seen in the P-bodies, smFISH probes in different parts of the transcripts could be used. Additionally, MBSV6 loops could be used for comparison (see below).

In response to the mentioned reports, Tutucci et al (PMID: 29131164) developed an improved MS2 system (MBSV6) with extended linkers and a reduced number of stem loops. This system has also been used in mammalian cells (PMID: 31407274). The authors should consider to update the MS2 loops used in RIDR to MBSV6 or a further optimised variant.

Minor comments:

Figure 1c: labels on x-axis seem to be incomplete (+/- in the first row).

mRNA decay kinetics: "... (U-2 OS) cells stably expressing both the mCherry-MBS reporter RNA and the RIDR construct, >90% of mCherry-MBS RNA disappeared upon induction for 2 hours" How does this relate to the KD by 74% achieved in Figure 1c? How long was treatment time in Figure 1c? Information on rapamycin concentration and treatment time should be provided for all experiments shown.

It is unclear which smFISH probes were used in the different experiments and what was used for untagged controls. The positions of the used smFISH probes in the transcripts should be indicated for all experiments

Typos:

line 97: "system system"

Response to reviewers

We would like to thank all the reviewers for their constructive comments and suggestions that helped us improve this manuscript. We have conducted extensive experiments according to reviewers' suggestions and performed additional analyses of existing data to enrich the manuscript. We have provided additional supplemental figures and revised the main manuscript. The major modifications have been highlighted in yellow.

Reviewer #1 (Remarks to the Author)

In the manuscript by Blake et al., the authors developed a rapid inducible RNA decay system, RIDR, which employs the MS2 system and the inducible tethering of SMG7C. Using RIDR, the authors showed that the tagged mRNA was targeted to the P-bodies where the mRNAs were degraded more rapidly than in the cytoplasm. RNA turnover is difficult to capture because of its stochastic nature. RIDR overcame this challenge by inducing synchronous RNA decay, thus is a useful tool to study RNA metabolism. The conclusions of this study were supported by well-designed experiments. The manuscript will be further improved when the following questions are addressed.

Major points:

1. In figure 3 and line 540-541, the authors built a quantitative model to estimate the RNA decay rate in P-bodies. The RNA synthesis rate was not taken into account. And "there were not enough features in the curves to unambiguously determine all parameters". To improve the model, the authors could perform additional experiments under the condition of no Rapa. First, the RNA decay rate in cytoplasm, k_{CT} , could be determined by inhibiting transcription and monitoring the RNA level over time without adding Rapa. Next, the RNA synthesis rate could be derived from the RNA steady state level when no Rapa was added. These additional experimental data will help more accurately determine the parameters in the model.

Response: Thank you for your suggestions to improve the model. To eliminate the confounding RNA synthesis, we have added the transcription inhibitor DRB at the same time as Rapamycin in all time-course experiments. We addressed this point on Page 7. There is another reason why we did not use the steady-state RNA decay value in the model as suggested by the reviewer: after adding rapamycin, the RNA decay in cytoplasm still occurs and may be faster than the steady-state value.

2. How do light stress and oxidative stress affect the decay of GAPDH and PolR2A mRNA?

Response: We thank the reviewer for raising this important question. To address this question, we conducted extensive control experiments. We performed smFISH-IF experiments to probe endogenous *mGAPDH* and *mPolR2A* mRNAs under light and oxidative stress conditions. To mimic the light stress in live cell imaging experiments, we used a commercial 24-well blue LED light stimulator (Live Cell Instruments). The light stress dosage can be controlled by varying the light intensity and exposure time. After adding Rapa or vehicle control with DRB, we incubated cells in the stimulator under light stimulation. To introduce oxidative stress, we treated cells with sodium arsenite. To summarize, cells were incubated under light or with sodium arsenite (200 μ M) stress for 30 minutes, treated with Rapamycin + DRB for 2 hours, then fixed. Then we conducted smFISH-IF with probes against *ACTB-MBS*, *mGAPDH*, *mPolR2A* RNAs and an antibody against DCP1a. Under light or oxidative stress condition, the RIDR induced *ACTB-MBS* decay was

significantly disrupted, as we have found previously. However, the decay of *mGAPDH* and *mPolR2A* RNAs was not significantly influenced. We have incorporated these new results in Supplemental Figures 9d-e.

3. In figure 5, the authors monitored the RNAs in P-bodies by measuring the Halo signal intensities. There are two populations of FKBP-Halo-MCP in the cells, the ones that are associated the MS2-tagged mRNA and the ones that are not. Do the latter population also move to P-bodies through the induced tethering of SMG7C? If so, the authors need to use an alternative approach, such as smFISH, to monitor RNA decay in P-bodies.

Response: We thank the reviewer for raising this important point. Another reviewer also had similar concerns. Therefore, we performed an additional control experiment. In the live-cell experiment, we relied on FKBP-HaloTag-tdMCP to detect the fate of mRNA. Indeed, if FRB-SMG7C was localized in P-bodies, FKBP-HaloTag-tdMCP could be recruited in P-bodies when treated with Rapa. To exclude this possibility, we performed an experiment to demonstrate that FRB-SMG7C did not colocalize with P-bodies. In agreement with the literature, we found that the full-length SMG7 was slightly enriched in P-bodies. These new control data are presented in Supplemental Figure 5. Furthermore, to directly answer the reviewer's request, *we have already used smFISH to probe RNA directly in fixed cell experiments* (Figs, 2, 3). The granules we observed were indeed RNA species, not just FKBP-HaloTag-tdMCP. To show the agreement between fixed-cell and live-cell measurements, we plotted the P-body RNA counts measured by both methods together (Supplemental Figure 9a), which showed similar trends. Therefore, we conclude that the accumulation of FKBP-HaloTag-tdMCP in P-bodies after Rapa is not due to pre-existing FRB-SMG7C in P-bodies, but instead due to decaying mRNAs that localize there.

Minor point:

Line 36, "specific a" should be "a specific".

Response: Thank you for attention to detail. The typo has been fixed.

Reviewer #2 (Remarks to the Author)

Blake et al., "A Rapid Inducible RNA Decay system reveals fast mRNA decay in P-bodies" In this manuscript, the authors describe a tool they have engineered to quantitate RNA degradation rates in living cells. The system consists of two halves which can rapidly be brought together in vivo by addition of rapamycin. The first half consists of an RNA decay factor (SMG7C) fused to the FKBP-Rapamycin binding domain (FRB). The other half of the pair is made out of FKBP fused with the RNA-binding MS2 coat protein (MCP), which recognizes MS2 binding sites in RNA. In the presence of rapamycin, FKBP rapidly binds to FRB, bringing the decay factor close to the mRNA and initiating its degradation. In addition, a target mRNA is labeled with MBS sequences where the MCP can bind, and the FKBP/MCP constructs includes a Halo tag to enable detection by fluorescence of the FKBP/MCP protein. This is put downstream of an mCherry tag, and so degradation of the mRNA induced by SMG7C being recruited there can be monitored in parallel by decreasing mCherry fluorescence.

This construct is used to study the decay rate of selected tagged mRNAs. It was found that mRNA levels could be reduced by 95 % within 2 h, which is considerably more efficient than RNAi. In the course of these observations, it was found that upon induction of RIDR, labeled RNA accumulates at P-bodies.

Quantification of RNA degradation within P-bodies and diffusely in the cytoplasm demonstrated that accelerated RNA decay, relative to the cytoplasm, takes place in P-bodies. The alternative possibility would be that labeled RNA migrated to P-bodies simply for storage.

Interestingly, it turns out that the estimated RNA degradation rates in P-bodies differ greatly between standard conditions and stress (allegedly oxidative stress). The signal duration of the labeled mRNA from P-bodies is greatly prolonged when the cells are exposed to high intensity laser, or arsenite. The authors argue that during stress, the physiology of PBs changes from rapid mRNA degradation to some other, yet undefined state. For a long time, there has been contradictory statements in the literature whether P-bodies are sites of RNA decay, as originally held, or whether they are repositories of decay factors and the majority of RNA degradation takes place outside of them in the cytoplasm. Indeed, it is conceivable that these new findings could reconcile these opposing views.

The experimental design in the paper is well conceived, and the results properly analyzed and interpreted. There are innovative and basic science aspects of this work. Although the RIDR system was custom built for this paper and underlie all the findings in it, it uses already established components for chemically inducible dimerization. I find the basic science findings more important here, as they shed light on fundamental issues of how P-bodies are related to RNA decay. This paper should have a significant impact and be of interest to a wide audience in the RNA biology field, and biomedical sciences at large. The writing style in the manuscript is well balanced and gives enough experimental detail without sacrificing overview. The rationale for designing the test system and performing the experiments are given with the proper background from previous work from other labs.

The selection of literature references is adequate, they cite most of the relevant publications in the field. Overall, this is a substantial contribution that deserves publication in a high-profile journal. There are however some issues that need to be addressed:

Response: We appreciate the detailed response and enthusiasm of the reviewer. We thank the reviewer's opinion about our contribution to the basic science of RNA decay.

General point:

1. SMG7, the mRNA decay factor used in the constructed system, is itself located in P-bodies (Unterholzener and Izaurralde 2004). Can this affect the localization of degrading mRNAs to P-bodies? The authors do not discuss this issue. In the same publication, it was shown that SMG6 does not localize to P-bodies but to separate granules. The authors of the present manuscript could rule this possibility in or out by testing their SMG6-based construct in parallel experiments as with SMG7.

Response: We thank the reviewer for raising this important point, which is worth further investigation. Reviewer 1 had a similar concern (Point 3). In the Unterholzner et al. Mol Cell 2004 paper, it was found that while the full length SMG7 protein (SMG7FL) colocalized with P-bodies, the C-terminal region of SMG7 (SMG7C) did not. To confirm this in our own construct, FRB-eGFP was fused to SMG7C and SMG7FL respectively to track their localization. We transiently transfected these two constructs into blank U-2 OS cells. We probed the eGFP signal in P-bodies, labeled by DDX6 immunofluorescence. SMG7C did not form noticeable puncta. However, in agreement with Unterholzner and colleagues, the SMG7FL formed small puncta colocalizing with P-bodies. These new data were presented in the Supplemental Figure 5.

To address the reviewer's quest about SMG6, we evaluated the SMG6PIN domain in the RIDR construct. We cloned a modified RIDR construct by substituting SMG7C with the SMG6PIN domain. Unfortunately,

our initial flow cytometry data indicated that the SMG6PIN did not have high knockdown efficiency after Rapa induction, even though it had moderate knockdown efficiency when directly tethered (Figure S1c). We shared this result with the reviewer in Figure R1. Further optimization of the construct is necessary to improve the RIDR efficiency with SMG6PIN, but might require substantial efforts. We believe it does not enrich our story and is outside the scope of this revision. We appreciate the reviewer's understanding of this point.

2. The nature of the engineered inducible degradation system presented here is such that the start of degradation is forced by addition of rapamycin, leading to the joining of the two CID components which brings the RNA decay factor SMG7C in vicinity of the mRNA. This means that the physiological regulation of degradation of the mRNA under study is short-circuited, and the events leading up to the degradation itself are eliminated. Could artifacts arise this way? Could degradation rates be overestimated because the decay machinery is running without the normally imposed constraints?

Response: We thank the reviewer for the comment and we completely agree with the reviewer's assessment. Just like many inducible systems (such as siRNA, protein degrons, etc), we intentionally bypassed "the physiological regulation of degradation" to achieve rapid RNA knockdown. The purpose of RIDR is not to measure the physiological decay rates, but rather to increase the degradation of a specific transcript. In short, the decay rate is artificially increased on purpose. To highlight this point, we added the following sentences in the discussion (Page 15): "The measured rate of decay in P-bodies was faster than in the cytoplasm (Fig. 3g). Note that these decay rates are likely faster than physiologically rates, as the decay is induced and upstream processes have been bypassed."

3. Further, can the addition of many copies of MBS sequences to the 3'-UTR and expression of the MS2 coat protein influence the results? The MCP can form aggregates when expressed at high levels, can they drag the attached mRNAs with them into punctate structures?

Response: We thank the reviewer for highlighting this important issue, which has been raised by another reviewer as well. We have addressed the point in the response to Reviewer 1 in point 3. To further comment on this issue, the concentration of MCP does not change before and after we add Rapa. We have shown that FKBP-HaloTag-tdMCP does not form aggregates or colocalized puncta with P-bodies in our live-cell imaging experiments, where there is no MCP aggregate until after Rapa is added (Figure 5). Furthermore, our additional supplemental data showed that FRB-SMG7C itself does not colocalize with P-bodies. Therefore, it is not P-body-localized FRB-SMG7C that "drags" MCP to form puncta. Instead, it is the decaying mRNAs that carry MCP into the P-bodies. To further clarify this point, we have added this discussion on Page 9.

4. Also, the inducible system relies on SMG7C for RNA degradation. SMG7 acts in the NMD pathway, downstream of e.g. Upf1 and SMG5. It was shown previously (Unterholzner and Izaurralde 2004) that SMG7 tethered to a transcript will execute its degradation, even if the premature termination codon normally triggering NMD is eliminated. The question arises if RNA-tethered SMG7 will degrade any mRNA, irrespective of whether that mRNA is a natural NMD target or not. To mutate a premature termination codon in a natural NMD target and show that SMG7 will still degrade it, does not prove that SMG7 will act on all mRNAs, or even on a large fraction of the mRNA population. The authors should discuss the generality of their system against this background.

Response: We thank the reviewer for pointing out the limitation of RIDR. We were actually inspired by this work of Unterholzner and Izaurralde et al. We completely agree with the reviewer and we are not claiming that SMG7C acts on all mRNAs, or even a large fraction of the mRNA population. To clarify this limitation, we added a sentence in the Discussion on Page 16: “It is not yet known whether tethering of SMG7C can cause decay of all mRNAs, or just MS2-tagged RNAs.”

Specific points:

Line 214: “decay rate in P-bodies was significantly higher than in the cytoplasm” Does this refer to the half-life of a particular mRNA species, or to the total degradation capacity of P-bodies (i.e. taking into account the total number of RNA molecules of a certain species)?

Response: We thank the reviewer for pointing out the ambiguity in our writing. We were referring to the decay rate of the specific mRNA targeted for decay. In this case, the specific RNA species being referred to was ACTB-MBS. To make this clear, we have edited the text on Page 11: “the fitting revealed that the decay rate of ACTB-MBS mRNA in P-bodies was significantly higher than that in the cytoplasm.”

Line 220: P-body constituent proteins are also present in the cytoplasm, so the result of depleting them will depend on their relative concentration in P-bodies or the cytoplasm.

Response: This is an excellent point. According to smFISH-IF, DDX6 and XRN1 were depleted in both P-bodies and cytoplasm after siRNA treatment (Supplemental Figure 8a and 8f). However, reliable quantification of the relative concentrations with immunofluorescence is challenging, especially given their diffusive signals in the cytoplasm. Therefore, we have edited the text to clarify this point on Page 11: “RNAi of XRN1 resulted in significant depletion of XRN1 protein both in the cytoplasm and P-bodies (Fig. S8a).” We also added the following to Page 12: “When DDX6 was knocked down with siRNA (Figs. S8f-g), there were no visible P-bodies (Fig. 4d) and reduced cytoplasm signals of DDX6 were observed (Fig. S8f).”

Line 226: The interpretation of this result depends on Xrn1 being enriched in P-bodies – but have the authors or anybody else shown that it is enriched?

Response: Thank you for pointing out this gap in our interpretation. Indeed, previous studies have shown that XRN1 is enriched in P-bodies (Sheth and Parker. *Science*. 2003; Kedersha et al. *J Cell Biol*. 2005). To confirm this ourselves, we performed a smFISH-IF experiment where we used a primary antibody against XRN1. We observed RNA granules colocalizing with XRN1 puncta. This new data is presented in Supplemental Figure 3e.

Line 260: Arsenite is a classical strong inducer of stress granules (oxidative stress by other agents is not a strong stress granule inducer). Arsenite has several effects on cells, not necessarily related to oxidative stress. With this in mind, is it possible that the prolonged signal of labeled mRNA in P-bodies is due to an increasing background under these conditions of stress granules, which are sometimes seen adjacent to P-bodies?

Response: We thank the reviewer for raising this important point about the interaction between P-bodies and stress granules. Previous studies have shown that stress granules and P-bodies do not colocalize, but come close to one another under stressed conditions. To investigate the effect of stress granules on RIDR, we performed immunofluorescence against G3BP1 (a stress granule marker) in ACTB-MBS MEF cells after treatment by 200µM sodium arsenite and induction with Rapamycin simultaneously. We found that stress

granules often localize adjacent to RNA granules after treatment with Rapa and arsenite stress but rarely colocalize (Supplemental Figure 3c). The *ACTB-MBS* RNA did not accumulate in stress granules in either No-Rapa or Rapa conditions (Supplemental Figure 3b-c, respectively).

We verified this in our live-cell imaging using the same cells in Figure 5, *ACTB-MBS* MEF stably expressing RIDR and eGFP-DDX6, for 2 hours after treatment with 200 μ M arsenite. We tracked the P-bodies using eGFP-DDX6 signal, measured the RNA signal in the P-bodies over time, and found that there was no increase of RNA signal in P-bodies when just NaAsO₂ was added (Supplemental Movie 5).

Line 281: Knocking down XRN1 leads to accumulation of labeled mRNA in P-bodies, which also persisted longer. But eliminating XRN1 will affect mRNA degradation globally, also in the cytoplasm. Maybe the labeled mRNA accumulates simply because its concentration in the cytoplasm also increases, and it has nowhere to go to be degraded?

Response: We thank the reviewer for the comments. Yes, eliminating XRN1 does influence global degradation, both in the P-bodies and in the cytoplasm. Indeed, we observed increased cytoplasmic mRNA when XRN1 was knocked down compared to non-target siRNA control throughout the duration of the RIDR experiment. Our model does not argue that P-bodies are the only place for *ACTB-MBS* mRNA to degrade. As the reviewer said: “the possibility that mRNA has nowhere to go and ends up in P-bodies” is very possible and does not contradict our model. In fact, the percentage of mRNA in the cytoplasm decreased in the XRN1 KD condition (Supplemental Figure 8e). We discussed this new point in the manuscript on Page 12: “Knocking down XRN1 results in a higher level of cytoplasmic mRNA than NC siRNA control after RIDR induction. However, the accumulation of mRNA in P-bodies is not simply due to increased cytoplasmic mRNA counts. In fact, the percentage of mRNA in the cytoplasm decreased in the XRN1 RNAi condition (Supplemental Figure S8e)”

Minor comments:

Line 75: “disappearance of signal being frequently confounded by imaging artifacts” Could the authors clarify what they mean here?

Response: During an imaging experiment, the disappearance of fluorescent signal might come from several reasons, for example, photobleaching, particles leaving the imaging focal plane or the field of view. To clarify this point, we edited the text to be “However, the spatiotemporal dynamics of RNA decay in cells remain poorly understood, due to the transient nature of degradation and the disappearance of signal being frequently confounded by imaging artifacts, such as photobleaching, diffusing out of focus or out of the field of view.”

Reviewer #3 (Remarks to the Author)

A Rapid Inducible RNA Decay system reveals fast mRNA decay in P-bodies by Blake et al.

Blake et al. present a rapid inducible system that allows to target a specific RNA of interest for RNA decay in living cells. The system builds on the recruitment of the C terminus of SMG7 via the MS2 coat protein (MCP) and the integration of MS2 stem loops (MBS) into the RNA of interest. Inducibility is achieved via the chemically inducible dimerization system which brings together SMG7C and MCP upon addition of rapamycin. The authors perform a series of experiments to establish the system and validate its

performance for both ectopically and endogenously expressed MS2-tagged RNAs of interest. Of note, they observe an accumulation of the tagged mRNAs in P-bodies. Following up on this observation, they test the impact of translation and different RNA decay factors. They also present a mathematical model to support that the trapping of RNAs in P-bodies allows for faster mRNA decay.

Overall, the authors introduce an elegant tool for the specific targeting and degradation of RNA molecules. They demonstrate that the system works very fast and efficient and allows for the observation of mRNA decay kinetics at the level of individual RNA molecules. Since single-molecule observations will be key to ultimately unravel RNA regulation in cells, I think that the developed tool will be a valuable asset for the community. Regarding the second part of the manuscript, my major concern is whether the observed P-body localisation could be influenced by the MS2 loops used to tag the RNAs of interest (see below).

Major points:

1. As the authors point out, rapamycin itself has an influence on translation which may in turn impact on mRNA decay. Can the authors quantitatively measure this effect? Could this relate to the ~10% Halo KD observed across conditions in Figure 1c? In addition, the minimal rapamycin concentration needed for efficient dimerisation should be established in titration curves.

Response: We thank the reviewer for raising this important point. Rapamycin concentration for all previous experiments in the paper was fixed to 100nM. To characterize the dependence of RIDR on the Rapamycin concentration, we performed a titration of the Rapamycin from 0.1 nM to 1000 nM. We found that the efficiency of induced RNA decay plateaued at 10nM Rapamycin. The degradation efficiency decreases at 1nM Rapamycin. This new data is presented in Supplementary Figure 1h.

In this context, Figure 1c is central to demonstrating the functionality of the RIDR system and the influence of its components. Yet, the information provided is not sufficient to evaluate the obtained results. It should be explained more clearly how the data were normalised and how the "% KD efficiency" values were obtained. Unnormalised data and FACS measurements for individual samples (as in Figure 1a) could be added in the Supplementary to allow for a full evaluation of the data.

Response: We thank the reviewer for the ideas to improve the data presentation. We adopted the reviewer's suggestion and included the raw flow cytometry data. In Figure 1b, an example FACS scatter plot was provided for each condition in Figure 1c. We added raw flow cytometry data for the supplemental experiments as well (Supplemental Figures 1d,g). We hope this shows the effect of SMG7C/RIDR more intuitively.

3. The authors use MBSV5 as MS2 loops for tagging the RNA of interest. However, multiple studies in yeast suggested that MS2 loops can interfere with 5'-to-3' degradation and produce RNA fragments that end up in P-bodies (PMID: 26092944, 12730603, 28096443, 27090788). This has also been explicitly shown for the MBSV5 loops that were used in the present study (PMID: 29131164). If this would also be the case in the system presented here, it would have strong confounding effects on the observations made.

Further investigations regarding this potential artifact are necessary to ensure the validity of the conclusions. To assess whether complete RNAs or decay fragments are seen in the P-bodies, smFISH probes in different parts of the transcripts could be used. Additionally, MBSV6 loops could be used for comparison (see below).

Response: We thank the reviewer for raising this important issue that necessitates a thorough response. In the ACTB-MBS MEF cells, the MS2 was the original version: MBSv1. In the mCherry-24xMBSv5 (Fig. 1), the MS2 was synonymously mutated to remove the repeatability (Wu et al, 2015). As the reviewer pointed out, multiple studies in YEAST have shown that MBS may interfere with 5'→3' degradation and produce RNA fragments, but these studies were limited to yeast. In mammalian cells, both MS2 and PP7 can be efficiently degraded (Horvathova, et al, Mol Cell 2017). For example, in our experiment, we did not observe RNA granules at steady-state conditions. To address the reviewer's concern explicitly, we performed the suggested control experiment: two-color smFISH targeting the *mACTB* open reading frame (ORF) and MBSv1 separately. After induction with Rapa (30min), we observed that the ORF probes also formed granules colocalizing with MBSv1 and P-bodies, indicating that RNAs in the P-bodies are not simply MBS remnant fragments. This new data was presented in Supplemental Figures 4a-b and a discussion was added to Pages 8. For the suggested MBSv6 experiment, please see our next response point.

In response to the mentioned reports, Tutucci et al (PMID: 29131164) developed an improved MS2 system (MBSV6) with extended linkers and a reduced number of stem loops. This system has also been used in mammalian cells (PMID: 31407274). The authors should consider to update the MS2 loops used in RIDR to MBSV6 or a further optimised variant.

Response: Thank you for the clear and specific suggestion to try the improved MBSv6 reporter with our RIDR system. We constructed *cmv-mCherry-24xMBSv6* as suggested. We transiently transfected the RIDR construct and *mCherry-24xMBSv6* into blank U-2 OS cells. After induction by Rapa for 30 minutes, we observed *mCherry-MBSv6* RNA granules colocalizing with P-bodies. Similarly, transiently co-transfected *mCherry-24xMBSv5* and RIDR also formed RNA granules colocalizing with P-bodies upon induction. The new data is shown in Supplemental Figure 4c-f. Together with the previous point, we hope we have demonstrated that the mRNAs accumulating in P-bodies are not just artifactual MBS fragments.

Minor comments:

Figure 1c: labels on x-axis seem to be incomplete (+/- in the first row).

Response: Thank you for this. We double-checked and did not see this issue ourselves. We suspect it could be a problem with the PDF viewer. For example, when using Microsoft Edge to view the figures, sometimes there were text portions removed, but Adobe always showed the text properly for us.

mRNA decay kinetics: "... (U-2 OS) cells stably expressing both the *mCherry-MBS* reporter RNA and the RIDR construct, >90% of *mCherry-MBS* RNA disappeared upon induction for 2 hours" How does this relate to the KD by 74% achieved in Figure 1c? How long was treatment time in Figure 1c? Information on rapamycin concentration and treatment time should be provided for all experiments shown.

Response: Thank you for this suggestion on improving the clarity of the paper. We have added the important information about the experiment to the legends of the figures, instead of only listing these important details in the Methods section. This includes the Rapamycin concentration (100nM) and the detailed times of transfection and Rapamycin induction (12-16 hours).

We praise the reviewer's attention to the detailed knock-down efficiency numbers. We suspect the discrepancy of the knockdown efficiency may come from the measurement targets. In Figure 1c, we quantified the knockdown by measuring mCherry protein levels with flow cytometry, while in Figure 1f, we measured mRNA levels directly. The mCherry protein is more stable than the *mCherry-MBS* mRNA. So mCherry protein created before RNA knockdown may persist even after the mRNA has been degraded. The reporter mCherry-24xMBSv5 and the RIDR plasmids were transiently transfected at the same time. It may take some time for RIDR to accumulate and knock down *mCherry-24xMBSv5* mRNAs. The expressed mCherry protein will accumulate even after the mRNA is degraded.

It is unclear which smFISH probes were used in the different experiments and what was used for untagged controls. The positions of the used smFISH probes in the transcripts should be indicated for all experiments

Response: Thank you for the suggestions to improve the data presentation. We have adapted the figures to show the FISH probe targeting site for each experiment, as well as live-cell imaging schematics.

Typos:

line 97: "system system"

Response: Thank you for pointing out this typo. It has been fixed.

Reviewer Figures

Figure R1: *SMG6PIN* does not perform well in the RIDR construct. **a)** Modified RIDR construct where *SMG6PIN* was cloned in place of *SMG7C* **b)** Raw Flow cytometry data of HEK293T cells transfected with FRB-*SMG6PIN*-IRES-FKBP-HaloTag-tdMCP-NLS and mCherry-24xMBSv5 reporter with 100nM Rapa (blue) and without Rapa (black) after ~16 hours of transfection. Rapamycin was added directly after transfection as in other experiments. **c)** Quantification of knockdown efficiency from flow cytometry data. Knockdown percentages are calculated by normalizing mCherry intensity in the + Rapa condition to the - Rapa condition.

Reviewers' Comments:

Reviewer #1:

Remarks to the Author:

The authors have addressed all my questions in the revised manuscript.

Reviewer #2:

Remarks to the Author:

The authors have adequately responded to all my questions and concerns. The specific points are listed below.

Reviewer #2 (Remarks to the Author)

Blake et al., "A Rapid Inducible RNA Decay system reveals fast mRNA decay in P-bodies"

In this manuscript, the authors describe a tool they have engineered to quantitate RNA degradation rates in living cells. The system consists of two halves which can rapidly be brought together in vivo by addition of rapamycin. The first half consists of an RNA decay factor (SMG7C) fused to the FKBP-Rapamycin binding domain (FRB). The other half of the pair is made out of FKBP fused with the RNA-binding MS2 coat protein MCP), which recognizes MS2 binding sites in RNA. In the presence of rapamycin, FKBP rapidly binds to FRB, bringing the decay factor close to the mRNA and initiating its degradation. In addition, a target mRNA is labeled with MBS sequences where the MCP can bind, and the FKBP/MCP constructs includes a Halo tag to enable detection by fluorescence of the FKBP/MCP protein. This is put downstream of an mCherry tag, and so degradation of the mRNA induced by SMG7C being recruited there can be monitored in parallel by decreasing mCherry fluorescence.

This construct is used to study the decay rate of selected tagged mRNAs. It was found that mRNA levels could be reduced by 95 % within 2 h, which is considerably more efficient than RNAi. In the course of these observations, it was found that upon induction of RIDR, labeled RNA accumulates at P-bodies. Quantification of RNA degradation within P-bodies and diffusely in the cytoplasm demonstrated that accelerated RNA decay, relative to the cytoplasm, takes place in P-bodies. The alternative possibility would be that labeled RNA migrated to P-bodies simply for storage. Interestingly, it turns out that the estimated RNA degradation rates in P-bodies differ greatly between standard conditions and stress (allegedly oxidative stress). The signal duration of the labeled mRNA from P-bodies is greatly prolonged when the cells are exposed to high intensity laser, or arsenite. The authors argue that during stress, the physiology of PBs changes from rapid mRNA degradation to some other, yet undefined state. For a long time, there has been contradictory statements in the literature whether P-bodies are sites of RNA decay, as originally held, or whether they are repositories of decay factors and the majority of RNA degradation takes place outside of them in the cytoplasm. Indeed, it is conceivable that these new findings could reconcile these opposing views.

The experimental design in the paper is well conceived, and the results properly analyzed and interpreted. There are innovative and basic science aspects of this work. Although the RIDR system was custom built for this paper and underlie all the findings in it, it uses already established components for chemically inducible dimerization. I find the basic science findings more important here, as they shed light on fundamental issues of how P-bodies are related to RNA decay. This paper should have a significant impact and be of interest to a wide audience in the RNA biology field, and biomedical sciences at large.

The writing style in the manuscript is well balanced and gives enough experimental detail without sacrificing overview. The rationale for designing the test system and performing the experiments are given with the proper background from previous work from other labs.

The selection of literature references is adequate, they cite most of the relevant publications in the field.

Overall, this is a substantial contribution that deserves publication in a high-profile journal. There are however some issues that need to be addressed:

Response: We appreciate the detailed response and enthusiasm of the reviewer. We thank the

reviewer's opinion about our contribution to the basic science of RNA decay.

General point:

1. SMG7, the mRNA decay factor used in the constructed system, is itself located in P-bodies (Unterholzner and Izaurralde 2004). Can this affect the localization of degrading mRNAs to P-bodies? The authors do not discuss this issue. In the same publication, it was shown that SMG6 does not localize to P-bodies but to separate granules. The authors of the present manuscript could rule this possibility in or out by testing their SMG6-based construct in parallel experiments as with SMG7.

Response: We thank the reviewer for raising this important point, which is worth further investigation. Reviewer 1 had a similar concern (Point 3). In the Unterholzner et al. Mol Cell 2004 paper, it was found that while the full length SMG7 protein (SMG7FL) colocalized with P-bodies, the C-terminal region of SMG7 (SMG7C) did not. To confirm this in our own construct, FRB-eGFP was fused to SMG7C and SMG7FL respectively to track their localization. We transiently transfected these two constructs into blank U-2 OS cells. We probed the eGFP signal in P-bodies, labeled by DDX6 immunofluorescence. SMG7C did not form noticeable puncta. However, in agreement with Unterholzner and colleagues, the SMG7FL formed small puncta colocalizing with P-bodies. These new data were presented in the Supplemental Figure 5.

- As SMG7C was mainly used in the present paper, showing that this construct does not localize to PBs is enough to take care of this concern.

To address the reviewer's quest about SMG6, we evaluated the SMG6PIN domain in the RIDR construct. We cloned a modified RIDR construct by substituting SMG7C with the SMG6PIN domain. Unfortunately, our initial flow cytometry data indicated that the SMG6PIN did not have high knockdown efficiency after Rapamycin induction, even though it had moderate knockdown efficiency when directly tethered (Figure S1c). We shared this result with the reviewer in Figure R1. Further optimization of the construct is necessary to improve the RIDR efficiency with SMG6PIN, but might require substantial efforts. We believe it does not enrich our story and is outside the scope of this revision. We appreciate the reviewer's understanding of this point.

- See above – my remark is no longer relevant if we know that SMG7C is not localized to PBs.

2. The nature of the engineered inducible degradation system presented here is such that the start of degradation is forced by addition of rapamycin, leading to the joining of the two CID components which brings the RNA decay factor SMG7C in vicinity of the mRNA. This means that the physiological regulation of degradation of the mRNA under study is short-circuited, and the events leading up to the degradation itself are eliminated. Could artifacts arise this way? Could degradation rates be overestimated because the decay machinery is running without the normally imposed constraints?

Response: We thank the reviewer for the comment and we completely agree with the reviewer's assessment. Just like many inducible systems (such as siRNA, protein degrons, etc), we intentionally bypassed "the physiological regulation of degradation" to achieve rapid RNA knockdown. The purpose of RIDR is not to measure the physiological decay rates, but rather to increase the degradation of a specific transcript. In short, the decay rate is artificially increased on purpose. To highlight this point, we added the following sentences in the discussion (Page 15): "The measured rate of decay in P-bodies was faster than in the cytoplasm (Fig. 3g). Note that these decay rates are likely faster than physiologically rates, as the decay is induced and upstream processes have been bypassed."

- This is an adequate response.

3. Further, can the addition of many copies of MBS sequences to the 3'-UTR and expression of the MS2 coat protein influence the results? The MCP can form aggregates when expressed at high levels, can they drag the attached mRNAs with them into punctate structures?

Response: We thank the reviewer for highlighting this important issue, which has been raised by

another reviewer as well. We have addressed the point in the response to Reviewer 1 in point 3. To further comment on this issue, the concentration of MCP does not change before and after we add Rapa. We have shown that FKBP-HaloTag-tdMCP does not form aggregates or colocalized puncta with P-bodies in our live-cell imaging experiments, where there is no MCP aggregate until after Rapa is added (Figure 5). Furthermore, our additional supplemental data showed that FRB-SMG7C itself does not colocalize with P-bodies. Therefore, it is not P-body-localized FRB-SMG7C that “drags” MCP to form puncta. Instead, it is the decaying mRNAs that carry MCP into the P-bodies. To further clarify this point, we have added this discussion on Page 9.

- The observation that FRB-SMG7C does not localize in PBs is important. Then formation of aggregates at high expression levels should not be a major concern.

4. Also, the inducible system relies on SMG7C for RNA degradation. SMG7 acts in the NMD pathway, downstream of e.g. Upf1 and SMG5. It was shown previously (Unterholzner and Izaurralde 2004) that SMG7 tethered to a transcript will execute its degradation, even if the premature termination codon normally triggering NMD is eliminated. The question arises if RNA-tethered SMG7 will degrade any mRNA, irrespective of whether that mRNA is a natural NMD target or not. To mutate a premature termination codon in a natural NMD target and show that SMG7 will still degrade it, does not prove that SMG7 will act on all mRNAs, or even on a large fraction of the mRNA population. The authors should discuss the generality of their system against this background.

Response: We thank the reviewer for pointing out the limitation of RIDR. We were actually inspired by this work of Unterholzner and Izaurralde et al. We completely agree with the reviewer and we are not claiming that SMG7C acts on all mRNAs, or even a large fraction of the mRNA population. To clarify this limitation, we added a sentence in the Discussion on Page 16: “It is not yet known whether tethering of SMG7C can cause decay of all mRNAs, or just MS2-tagged RNAs.”

- OK

Specific points:

Line 214: “decay rate in P-bodies was significantly higher than in the cytoplasm” Does this refer to the half-life of a particular mRNA species, or to the total degradation capacity of P-bodies (i.e. taking into account the total number of RNA molecules of a certain species)?

Response: We thank the reviewer for pointing out the ambiguity in our writing. We were referring to the decay rate of the specific mRNA targeted for decay. In this case, the specific RNA species being referred to was ACTB-MBS. To make this clear, we have edited the text on Page 11: “the fitting revealed that the decay rate of ACTB-MBS mRNA in P-bodies was significantly higher than that in the cytoplasm.”

- OK

Line 220: P-body constituent proteins are also present in the cytoplasm, so the result of depleting them will depend on their relative concentration in P-bodies or the cytoplasm.

Response: This is an excellent point. According to smFISH-IF, DDX6 and XRN1 were depleted in both P-bodies and cytoplasm after siRNA treatment (Supplemental Figure 8a and 8f). However, reliable quantification of the relative concentrations with immunofluorescence is challenging, especially given their diffusive signals in the cytoplasm. Therefore, we have edited the text to clarify this point on Page 11: “RNAi of XRN1 resulted in significant depletion of XRN1 protein both in the cytoplasm and P-bodies (Fig. S8a).” We also added the following to Page 12: “When DDX6 was knocked down with siRNA (Figs. S8f-g), there were no visible P-bodies (Fig. 4d) and reduced cytoplasm signals of DDX6 were observed (Fig. S8f).”

- There is an ambiguity as to what we should expect from lowering the concentrations of these proteins. I agree that quantitating protein concentrations in the different cellular compartments is very difficult. As long as this is acknowledged, as it is now in the revised text, this is enough.

- Line 226: The interpretation of this result depends on Xrn1 being enriched in P-bodies – but have the authors or anybody else shown that it is enriched?

Response: Thank you for pointing out this gap in our interpretation. Indeed, previous studies have shown that XRN1 is enriched in P-bodies (Sheth and Parker. Science. 2003; Kedersha et al. J Cell Biol. 2005). To confirm this ourselves, we performed a smFISH-IF experiment where we used a primary antibody against XRN1. We observed RNA granules colocalizing with XRN1 puncta. This new data is presented in Supplemental Figure 3e.

- This new piece of data, in line with earlier publications, is convincing.

Line 260: Arsenite is a classical strong inducer of stress granules (oxidative stress by other agents is not a strong stress granule inducer). Arsenite has several effects on cells, not necessarily related to oxidative stress. With this in mind, is it possible that the prolonged signal of labeled mRNA in P-bodies is due to an increasing background under these conditions of stress granules, which are sometimes seen adjacent to P-bodies?

Response: We thank the reviewer for raising this important point about the interaction between P-bodies and stress granules. Previous studies have shown that stress granules and P-bodies do not colocalize, but come close to one another under stressed conditions. To investigate the effect of stress granules on RIDR, we performed immunofluorescence against G3BP1 (a stress granule marker) in ACTB-MBS MEF cells after treatment by 200 μ M sodium arsenite and induction with Rapamycin simultaneously. We found that stress granules often localize adjacent to RNA granules after treatment with Rapa and arsenite stress but rarely colocalize (Supplemental Figure 3c). The ACTB-MBS RNA did not accumulate in stress granules in either No-Rapa or Rapa conditions (Supplemental Figure 3b-c, respectively).

We verified this in our live-cell imaging using the same cells in Figure 5, ACTB-MBS MEF stably expressing RIDR and eGFP-DDX6, for 2 hours after treatment with 200 μ M arsenite. We tracked the P-bodies using eGFP-DDX6 signal, measured the RNA signal in the P-bodies over time, and found that there was no increase of RNA signal in P-bodies when just NaAsO₂ was added (Supplemental Movie 5).

- These new experiments show that even if there is a physical close contact with SGs with arsenite, it is not enough to influence the signal measured by the authors.

Line 281: Knocking down XRN1 leads to accumulation of labeled mRNA in P-bodies, which also persisted longer. But eliminating XRN1 will affect mRNA degradation globally, also in the cytoplasm. Maybe the labeled mRNA accumulates simply because its concentration in the cytoplasm also increases, and it has nowhere to go to be degraded?

Response: We thank the reviewer for the comments. Yes, eliminating XRN1 does influence global degradation, both in the P-bodies and in the cytoplasm. Indeed, we observed increased cytoplasmic mRNA when XRN1 was knocked down compared to non-target siRNA control throughout the duration of the RIDR experiment. Our model does not argue that P-bodies are the only place for ACTB-MBS mRNA to degrade. As the reviewer said: “the possibility that mRNA has nowhere to go and ends up in P-bodies” is very possible and does not contradict our model. In fact, the percentage of mRNA in the cytoplasm decreased in the XRN1 KD condition (Supplemental Figure 8e). We discussed this new point in the manuscript on Page 12: “Knocking down XRN1 results in a higher level of cytoplasmic mRNA than NC siRNA control after RIDR induction. However, the accumulation of mRNA in P-bodies is not simply due to increased cytoplasmic mRNA counts. In fact, the percentage of mRNA in the cytoplasm decreased in the XRN1 RNAi condition (Supplemental Figure S8e)”

- At least there is no increase of the ACTB-MBS mRNA in the cytoplasm after addition of rapamycin, so then this should not be a driver of accumulation in PBs.

Minor comments:

Line 75: "disappearance of signal being frequently confounded by imaging artifacts" Could the authors clarify what they mean here?

Response: During an imaging experiment, the disappearance of fluorescent signal might come from several reasons, for example, photobleaching, particles leaving the imaging focal plane or the field of view. To clarify this point, we edited the text to be "However, the spatiotemporal dynamics of RNA decay in cells remain poorly understood, due to the transient nature of degradation and the disappearance of signal being frequently confounded by imaging artifacts, such as photobleaching, diffusing out of focus or out of the field of view."

- OK

Reviewer #3:

Remarks to the Author:

The authors fully addressed all my comments. Congratulations to this well-designed study which introduces an elegant new tool and provides interesting new insights into RNA decay mechanisms in cells.